# Mic19 depletion impairs endoplasmic reticulum-mitochondrial contacts and mitochondrial lipid metabolism and triggers liver disease

Jun Dong[1,7], Li Chen [1,2,7], Fei Ye[1,7], Junhui Tang[1], Bing Liu[1], Jiacheng Lin[1], Pang-Hu Zhou[1], Bin Lu[3], Min Wu [1], Jia-Hong Lu [4], Jing-Jing He[5], Simone Engelender[6], Qingtao Meng[1], Zhiyin Song [1,2] ✉ & He He [1,2] ✉

Endoplasmic reticulum (ER)-mitochondria contacts are critical for the regulation of lipid transport, synthesis, and metabolism. However, the molecular mechanism and physiological function of endoplasmic reticulum-mitochondrial contacts remain unclear. Here, we show that Mic19, a key subunit of MICOS (mitochondrial contact site and cristae organizing system) complex, regulates ER-mitochondria contacts by the EMC2-SLC25A46-Mic19 axis. Mic19 liver specific knockout (LKO) leads to the reduction of ER-mitochondrial contacts, mitochondrial lipid metabolism disorder, disorganization of mitochondrial cristae and mitochondrial unfolded protein stress response in mouse hepatocytes, impairing liver mitochondrial fatty acid β-oxidation and lipid metabolism, which may spontaneously trigger nonalcoholic steatohepatitis (NASH) and liver fibrosis in mice. Whereas, the re-expression of Mic19 in Mic19 LKO hepatocytes blocks the development of liver disease in mice. In addition, Mic19 overexpression suppresses MCD-induced fatty liver disease. Thus, our findings uncover the EMC2-SLC25A46-Mic19 axis as a pathway regulating ER-mitochondria contacts, and reveal that impairment of ER-mitochondria contacts may be a mechanism associated with the development of NASH and liver fibrosis.

Mitochondrion is a dynamic organelle, it plays an important role in various cellular processes, including ATP production, lipid homeostasis, reactive oxygen species (ROS) generation, apoptosis, and cell signaling, etc.[1]. Mitochondria are enveloped by double membranes: the outer mitochondrial membrane (OMM) and the inner mitochondrial membrane (IMM). The IMM is composed of the inner boundary membrane (IBM) and the cristae membranes. IBM is adjacent to OMM, and the cristae membranes are invaginations of the IMM. The connections between IBM and cristae membranes are cristae junctions (CJs). Mitochondrial cristae and CJs are essential for mitochondrial morphology, structure, and functions[2–4].

Endoplasmic reticulum (ER) plays an important role in calcium homeostasis and synthesis of proteins and lipids[5]. ER-mitochondria contacts are vital to orchestrate cell biological process, such as calcium signal, phospholipid synthesis, and translocation[6,7]. The membranes which represent the physical association between the ER and mitochondria are called mitochondria-associated ER membranes (MAMs). Interestingly, MAMs are enriched in enzymes involved in lipid

biosynthesis[8]. Mitochondrial phospholipids, including phosphatidylcholine (PC), phosphatidylethanolamine (PE), cardiolipin (CL), phosphatidic acid (PA), phosphatidylserine (PS), and phosphatidylglycerol (PG), are critical for mitochondrial membrane organization and function[8,9]. PA and PS are synthesized in ER and can be imported into the OMM by MAMs for the synthesis of CL and PE, respectively[8,10]. In addition, PE and CL are required for integrity of mitochondrial membrane structures[11], and CL interacts with various mitochondrial proteins and is sensitive to oxidative stress[12]. However, how lipids are affected by the reduction of ER-mitochondria contacts and its physiologic significance in liver disease remains largely unknown. Nonalcoholic fatty liver disease (NAFLD), the most predominant chronic liver disease, affects approximately 30% of population worldwide[13]. NAFLD is a progressive clinical pathology ranging from steatosis to nonalcoholic steatohepatitis (NASH), hepatic fibrosis, cirrhosis, ultimately, hepatocellular carcinoma[14]. Steatosis is the phenomenon of simple triglyceride (TG) accumulation in hepatocytes with no harmful effects on liver functions. NASH is characterized by inflammation and steatosis[15]. However, hepatic fibrosis is accompanied by inflammation, aberrant deposition of extracellular matrix (ECM) and progress to fibrous scar[16,17]. Insulin resistance, oxidative stress, and inflammation have been considered as the major factors in the progress of NAFLD[18]. Recently, increasing evidence suggests that mitochondrial phospholipids homeostasis plays a vital role in the physiopathology of NAFLD[19,20]. Therefore, disorder of mitochondrial phospholipids metabolism may be a hallmark of NAFLD, but the underlying mechanism remains largely unknown.

MICOS complex, a multi-subunit complex of proteins, is verified to stabilize the structure of IMM and mitochondrial homeostasis[21]. Studies in yeast and mammals have identified several subunits of MICOS complex, including Mic10, Mic13, Mic19, Mic25, Mic26, Mic27, and Mic60[22–24]. MICOS interacts with respiratory chain complexes, protein import machinery and cardiolipin[25–27]. Mic60 can interact with TOM complex and SAM (the sorting and assembly machinery) complex. Specifically, the Mic60-Mic19 subcomplex is thought to connect CJs to the OMM[28]. In addition, MICOS complex is associated with PS transport and assists the synthesis of mitochondrial PE[29,30]. Loss of MICOS complex subunit leads to mitochondrial fragmentation, CJs loss and reduced mitochondrial respiration[3,31]. Functionally, loss or mutations of MICOS complex subunits are associated with a series of diseases, including Parkinson's disease, cardiovascular disease, diabetes, liver disease and cancer[21,32,33]. However, the pathological mechanisms of abnormal MICOS leading to these diseases remains largely obscure.

Mic19 is one of the core subunits of MICOS complex and determines the stability of MICOS complex. We previously showed that Mic19 mediates the mitochondrial outer and inner membrane contacts by the Sam50-Mic19-Mic60 axis[34]. Mic19 deletion leads to mitochondrial ultrastructure changes including abnormal mitochondrial cristae abnormalities and loss of CJs, thus impairing ATP production[3]. However, the physiological role of Mic19 is still unknown. Here, we show that Mic19 regulates ER-mitochondria contacts via the EMC2-SLC25A46-Mic19 axis. Mic19 depletion caused mitochondrial phospholipids disorder, further leading to the structural abnormalities and stress of mitochondria and ER. Furthermore, Mic19 liver specific knockout (LKO) triggers NASH and liver fibrosis in mice. Whereas, re-expression of Mic19 in Mic19 LKO mouse liver dramatically ameliorates the phenotype of liver diseases. In summary, our findings demonstrate that Mic19 participates in mitochondrial lipid metabolism, and Mic19 dysfunction is highly involved in the occurrence and development of liver diseases in vivo.

## Results

### Mic19 is involved in the regulation of ER-mitochondria contacts
Mitochondria can contact with the ER to regulate vital cellular homeostatic functions. MAMs, the physical association between the ER and mitochondria, are essential for cellular homeostasis. Monteiro-Cardoso et.al recently reported that the lipid transfer proteins ORP5 and ORP8 were mainly located at MAM subdomains and physically linked to the MICOS complexes[30]. Mic19, a key subunit of MICOS complex, is significant for the integrity of MICOS and SAM complex[34]. We then investigated the role of Mic19 in mitochondria-ER contacts. HIS-SIM analysis revealed that co-localization of ER and mitochondria was reduced in Mic19 knockdown (KD) or knockout (KO) HeLa or COS7 cells (Fig. 1a–d, S1a–d). Furthermore, transmission electron microscopy (TEM) analysis showed that Mic19 KO caused mitochondrial ultrastructure abnormalities including loss of mitochondrial cristae junctions (CJs) and change of mitochondrial cristae arrangements but not change of mitochondrial length (Fig. 1e–g). Moreover, Mic19 KO led to a significantly increase of distance between ER and mitochondria, and caused a decrease of number of ER-mitochondria contacts (Fig. 1h, i), indicating that Mic19 depletion decreases ER-mitochondria contacts. Additionally, we investigated the effect on Mic19 depletion on mitochondrial number and content, which may impair ER-mitochondria contacts. HIS-SIM imaging and Western blotting analysis showed that mitochondrial number and contents (mitochondrial marker proteins including Tom20, Tom40, Tim23, SDHA and Cox4) was not changed in Mic19 KO cells (Fig. S1e–g). Furthermore, mito-keima assay displayed that mitophagy was not changed in Mic19 KO cells compared to control cells (Fig. S1h). Therefore, these results suggest that Mic19 participates in the regulation of ER-mitochondria contacts.

### Mic19 regulates ER-mitochondria contacts through the EMC2-SLC25A46-Mic19 axis
SLC25A46, a mitochondrial outer membrane protein, interacts with the MICOS complex and maintains interaction with the ER by EMC2[35]. Therefore, we investigated whether Mic19-regulated crosstalk between ER and mitochondria is related to SLC25A46. Co-immunoprecipitation (co-IP) analysis by using anti-Flag or anti-SLC25A46 antibodies showed that Flag-SLC25A46-coupled beads or SLC25A46-coupled beads but not control beads could precipitate Mic19, Mic60 and EMC2, indicating that SLC25A46 can interact with Mic19, Mic60 and EMC2 (Fig. 2a, b). Moreover, Mic19 KO decreased the protein level of SLC25A46 in cells (Fig. 2c, d), and SLC25A46 KD also led to the slight reduction of Mic19 protein (Fig. 2e, f). In addition, MG132 treatment could not inhibit CHX (cycloheximide, protein synthesis inhibitor)-induced reduction (degradation) of SLC25A46 in control and Mic19 KO cells (Fig. S2a–d), indicating that Mic19 KO-caused degradation of SLC25A46 is independent on the ubiquitin-proteasome pathway. Then, we explored the effect of mitochondrial proteases on Mic19 KO-induced the degradation of SLC25A46. Western blotting analysis revealed that the depletion of OMA1 or Yme1L led to significant inhibition of SLC25A46 degradation in Mic19 KO cells (Fig. S2e and S2f), indicating that mitochondrial protease OMA1 and Yme1L contribute to Mic19 KO-caused the degradation of SLC25A46. Also, Western Blotting analysis showed that CLS1 knockdown decreased the protein levels of Mic19, Mic60, and SLC25A46 in cells (Fig. S2g and S2h), suggesting that the degradation of SLC25A46 and MICOS subunits is probably cardiolipin-dependent.

Additionally, HIS-SIM imaging showed that SLC25A46 KD remarkably decreased ER-mitochondria contacts in HeLa cells (Fig. 2g, h). Furthermore, TEM analysis displayed that SLC25A46 KD caused the reduction of CJs in HeLa cells but not change mitochondrial length (Fig. 2i–k); moreover, SLC25A46 KD significantly increased the distance between ER and mitochondria, and reduced the number of mitochondria-ER contacts (Fig. 2l, m), indicating that SLC25A46 regulates ER-mitochondria contacts. It should be noted that mitophagy was not changed in SLC25A46 KD cells by mito-keima assay (Fig. S1i). Thus, these results revealed that Mic19 interacts with SLC25A46 to form EMC2-SLC25A46-Mic19 axis that contributes to the regulation of ER-mitochondria contacts.

## Mic19 deletion induces the disorder of mitochondrial lipid metabolism

ER-mitochondria contacts are critical for phospholipids transport between ER and mitochondria, regulating mitochondrial lipids metabolism[36,37]. We generated *Mic19* liver-specific knockout (LKO)

mice by crossing Mic19[flox/flox] with Albumin-Cre transgenic mice that specifically express Cre in mouse hepatocytes (Fig. S3a). Western blotting analysis showed that Mic19 was specifically depleted in mouse liver but was expressed in other mouse organs including muscle, heart, spleen, kidney, and brain (Fig. S3b). Then, we isolated mitochondrial

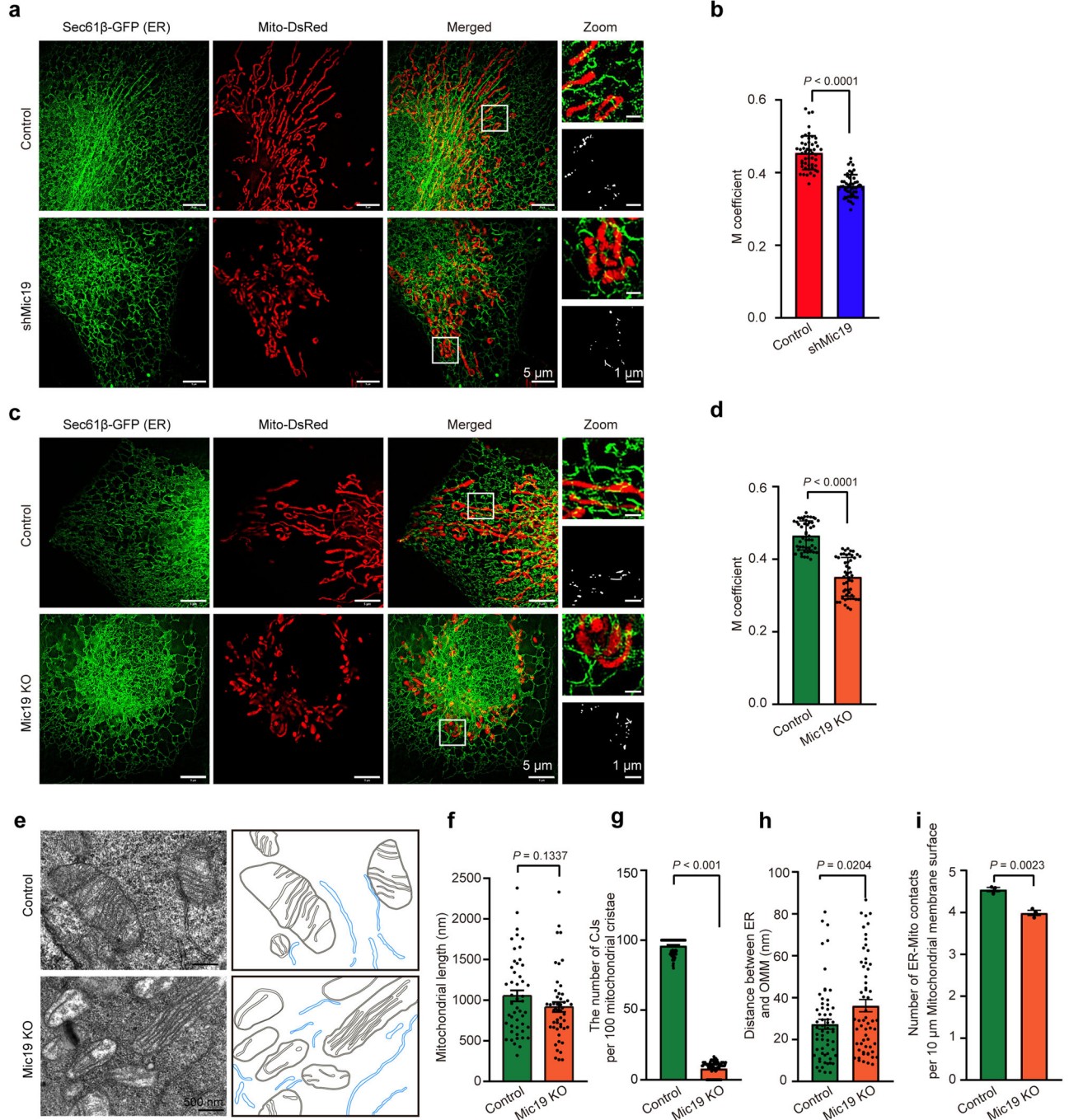

**Fig. 1 | Mic19 depletion decreases ER-mitochondria contacts. a, b** HeLa cells stably expressing mito-DsRed (a mitochondrial marker, red) and Sec61β-GFP (an ER marker, green) were infected with control (scrambled shRNA) or shMic19 lentiviral particles. 5 days later, these cells were imaged by HIS-SIM (High Sensitivity Structured Illumination Microscope) to analyze the co-localization of mitochondria and ER **a**. Co-localization of ER and mitochondria in "**b**" was further quantified by ImageJ software (**b**). *n* = 50 mitochondria examined over 3 independent experiments. **c, d** Control and Mic19 KO HeLa cells stably expressing Mito-DsRed and Sec61β-GFP were analyzed as described in "**a**" and "**b**". *n* = 50 mitochondria examined over 3 independent experiments. **e–i** ER-mitochondria contacts in control or Mic19 KO

HeLa cells were analyzed by transmission electron microscopy (TEM). Insets are mitochondrial membrane (dark gray) and ER membrane (blue) annotated with different colors (**e**). The mitochondrial length (**f**, *n* = 50 mitochondria), the number of cristae junctions (CJs) per 100 mitochondrial cristae (**g**, *n* = 100 mitochondria cristae), the distance between ER and OMM (**h**, *n* = 60 mitochondria) and the number of ER-Mito contacts per 10 μm mitochondrial membrane surface **i** were further analyzed by ImageJ software (*n* = 3 independent experiments). Data in **b**, **d** are presented as mean ± SD. Data in **f–i** are presented as mean ± SEM. Statistical significance was assessed by two-tailed Student's t-test. *P* values are indicated in the figure. Source data are provided as a Source Data file.

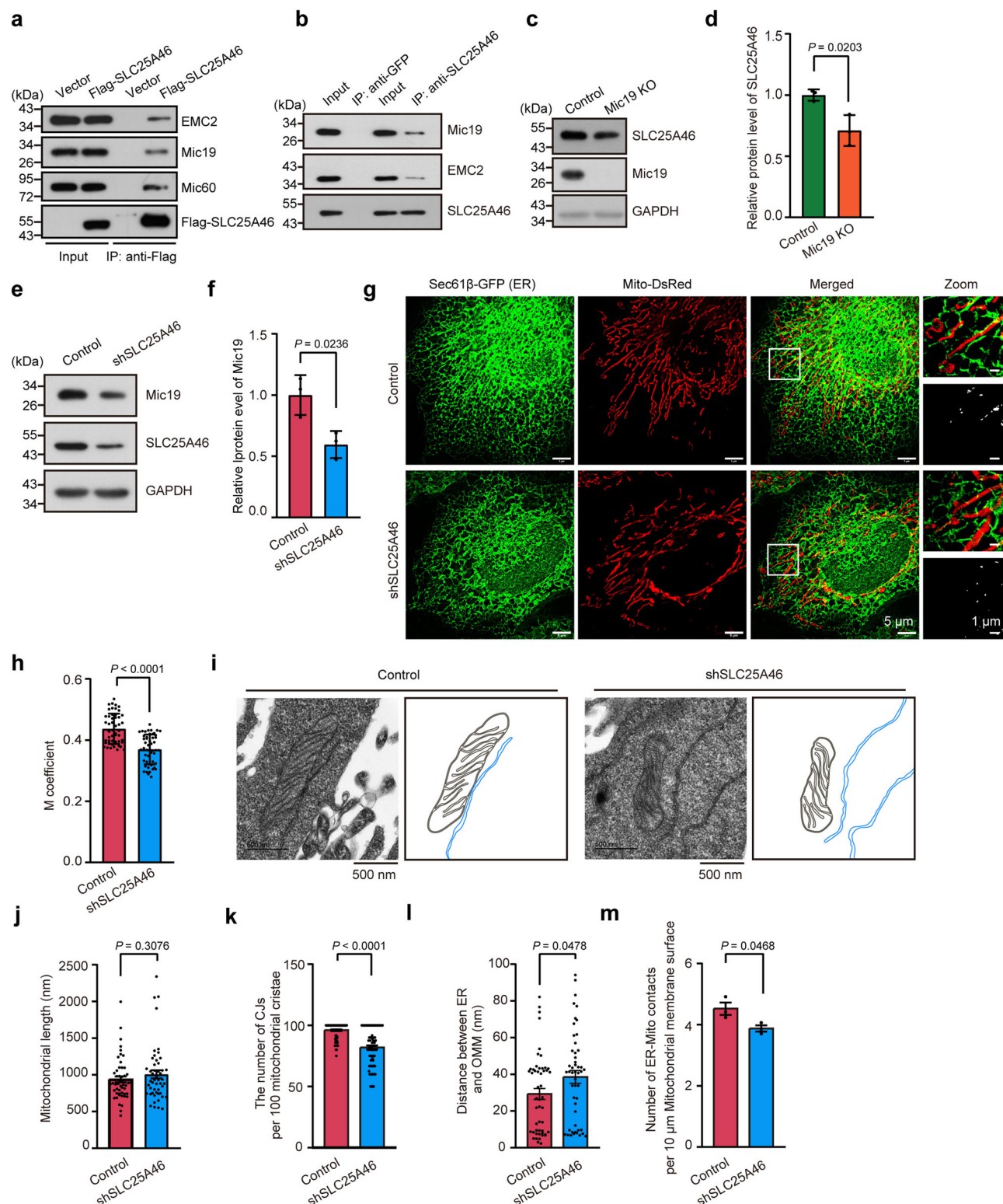

fraction and analyzed the mitochondrial phospholipid content in liver from 3-month-old Mic19 LKO mice. Lipidomics analysis revealed that the mitochondrial fraction of Mic19 LKO mouse liver had a decreased abundance of phosphatidic acid (PA), cardiolipin (CL), phosphatidylcholine (PC), phosphatidylserine (PS), and phosphatidylethanolamine (PE) (Fig. 3a, b), indicating that Mic19 depletion impairs the metabolism of mitochondrial phospholipids. CL is a phospholipid unique to mitochondria and mainly located in the IMM[38], and is critical for the maintenance of mitochondrial structure and function. Since the

EMC2-SLC25A46-Mic19 axis regulates mitochondria-ER contacts (Figs.1 and 2), we investigated the role of the EMC2-SLC25A46-Mic19 axis in CL metabolism. We isolated mitochondrial fraction of cells, and the purification of mitochondrial fraction were analyzed by Western blotting (Fig. S3c). Further lipidomics analysis showed that the CL level was decreased in the mitochondria fractions of Mic19 KO or SLC25A46 KD cells (Fig. 3c–f). In addition, we analyzed the level of CL by 10-N-Nonyl acridine orange (NAO, a cardiolipin binding dye) staining in control, Mic19 KO, and SLC25A46 KD cells. Confocal imaging and flow

**Fig. 2 | Mic19 is involved in the interaction between mitochondria and ER by EMC2-SLC25A46-Mic19 axis. a** 293 T cells were transiently transfected with empty vector (control) or vector coding for Flag-SLC25A46. After 36 h transfection, cell lysates were used for co-immunoprecipitation with anti-Flag M2 affinity gel at 4 °C overnight. Immunoblot analysis with antibodies against Flag, SLC25A46, Mic19, Mic60, or EMC2 (n = 3 independent experiments). **b** Liver lysates were used for co-immunoprecipitation with anti-GFP or SLC25A46 at 4 °C overnight, followed by immunoblot analysis with antibodies against Mic19, SLC25A46 or EMC2 (n = 3 independent experiments). **c–f** Control and Mic19 KO (**c**) or control and shSLC25A46 (**d**) HeLa cell lysates were analyzed by Western blotting with indicated antibodies. Representative immunoblots were from n = 3 independent experiments. Relative SLC25A46 (**d**) or Mic19 (**f**) protein levels were further evaluated by densitometry analysis using ImageJ software. **g, h** HeLa cells stably expressing Mito-DsRed (a mitochondrial marker, red) and Sec61β-GFP (an ER marker, green) were infected with control (scrambled shRNA) or shSLC25A46 lentiviral particles. 5 days later, cells were imaged by HIS-SIM to analyze the co-localization between mitochondria and ER (**g**). Co-localization (Manders coefficient) of ER and mitochondria was further quantified by ImageJ software (**h**). n = 50 mitochondria examined over 3 independent experiments. **i–m** ER-mitochondria contacts in control or shSLC25A46 HeLa cells are analyzed by transmission electron microscopy (TEM) (**i**). Insets are mitochondrial membrane (dark gray) and ER membrane (blue). The mitochondrial length (**j**, n = 50), the number of cristae junctions (CJs) per 100 mitochondrial cristae (30 mitochondria were analyzed) (**k**, n = 92 mitochondrial cristae), the distance between ER and OMM (**l**, n = 50) and the number of ER-Mito contacts per 10 μm mitochondrial membrane surface (**m**, n = 3) were further analyzed by ImageJ software (n = 3 independent experiments). Data in **d**, **f**, and **h** are presented as mean ± SD. Data in **j–m** are presented as mean ± SEM, statistical significance was assessed by two-tailed Student's t-test. P values are indicated in the figure. Source data are provided as a Source Data file.

cytometry analysis showed that Mic19 or SLC25A46 depletion caused a remarkable reduction of CL (Figs.3g–j and S3d-S3g), suggesting that loss of Mic19 or SLC25A46 impairs CL metabolism. Moreover, TMRM staining and flow cytometry analysis revealed that Mic19 KO and SLC25A46 KD decreased mitochondrial membrane potential in cells (Fig. S3h and S3i), which is probably due to impaired CL metabolism. In addition, Western blotting analysis revealed that the protein levels of CLS1 (cardiolipin synthase 1) and Tafazzin (TAZ, catalyzes transacylation to form mature cardiolipin) were not changed in Mic19 KO or SLC25A46 KD cells (Fig. S3j–S3m). Furthermore, the protein level of CLS1 or TAZ also remained unchanged in Mic19 LKO mouse liver (Fig.S3n and S3o). These data indicate that loss of Mic19 or SLC25A46 does not affect the ability of CL synthesis in mitochondria. It should be noted that although the protein levels of CLS1 and TAZ were unchanged, many other factors might lead to a reduction in cardiolipin level: the activity of phospholipid synthase may be inhibited, including enzyme inactivation modifications; additionally, substrate limitation may be a possible factor. Therefore, Mic19 deletion induces the disorder of mitochondrial lipid metabolism through EMC2-SLC25A46-Mic19 axis.

Together, the EMC2-SLC25A46-Mic19 axis regulates ER-mitochondria contacts and participates in mitochondrial lipid metabolism.

## Mic19 knockout leads to mitochondrial membrane disorganization and UPRmt in mouse liver

ER-mitochondria contacts are critical for the regulation of a variety of mitochondrial functions. To investigate the biological and physiological functions of EMC2-SLC25A46-Mic19 axis–mediated ER-mitochondria contacts, we generated Mic19 LKO mice (Fig. S3a). Mic19 LKO caused remarkable reduction of MICOS complex subunits including Mic10, Mic60, and Mic13, and led to a significant decrease of SLC25A46 (Fig. 4a, b). Since Mic19 depletion impairs the production of CL (Fig. 3g, h, S3d, e), which is critical for mitochondrial membrane organization and links oxidative stress and mitochondrial dysfunction[39,40], we assessed the effect of Mic19 LKO on mitochondrial membrane ultrastructure in mouse liver. TEM analysis showed that compared to control (Mic19flox/flox mouse liver cells), Mic19 LKO mouse liver cells displayed the increased distance between ER and mitochondria and the decreased number of ER-mitochondria contacts (Fig. 4c–e), confirming that Mic19 LKO impairs ER-mitochondria contacts. In addition, Mic19 LKO mouse liver cells showed significant reduction of mitochondrial cristae, dramatic loss of CJs (Fig. 4c, f, g); moreover, Mic19 LKO mouse liver mitochondrial cristae membrane was dramatically disorganized (Fig. 4c). Therefore, Mic19 LKO causes mitochondrial membranes remodeling in mouse liver, especially the disorder of mitochondrial cristae membrane.

Mitochondrial cristae are the main site for mitochondrial oxidative phosphorylation, which is critical for cellular energy production.

We then assessed the effect of Mic19 LKO-induced cristae disorganization on mitochondrial oxidative phosphorylation by performing Blue Native-PAGE analysis of mitochondria isolated from mouse livers. Strikingly, the levels of complex I, complex III, complex IV and complex V but not complex II were significantly decreased in Mic19 LKO mouse liver mitochondria (Fig. S4a and S4b). Next, we analyzed oxygen consumption rates of 3-month-old Mic19flox/flox and Mic19 LKO mouse liver cells by high-resolution respirometry with Oroboros O2k system. The oxygen consumption of Mic19 LKO mouse liver was consistently lower than that of Mic19flox/flox mouse liver under the treatment of various mitochondrial complex inhibitors (Fig. 4h and S4c-S4h), indicating that Mic19 LKO caused the defect of mitochondrial oxidative phosphorylation in mouse liver. Furthermore, Mic19 LKO mouse liver showed decreased ATP production and increased ROS level (Fig. S4i–S4k), suggesting that Mic19 LKO-induced mitochondrial membrane disorganization causes mitochondrial stress or dysfunction. The mitochondrial unfolded protein response (UPRmt) is activated during mitochondrial stress or dysfunction leading to the transcriptional up-regulation of protective genes including mitochondrial proteases and chaperones[41]. Thus, we investigated whether Mic19 LKO trigger the UPRmt in mouse livers. Western blotting analysis showed that the protein levels of UPRmt–related proteins including mitochondrial proteases (LONP1, ClpP), mitochondrial chaperone HSP60, and SOD2 (a surrogate marker of the UPRmt), but not Tom40 (mitochondrial marker) were remarkably enhanced in Mic19 LKO mouse liver (Fig. 4i, j), indicating that Mic19 LKO leads to the UPRmt in vivo.

In addition, we also assessed the effect of Mic19 LKO on ER homeostasis in mouse liver. The effect of Mic19 LKO on ER stress was tested. We determined the expression of the multiple markers including GRP78 (an integral ER stress sensor), Atf6 (a regulator of endoplasmic reticulum homeostasis), p-eIF2α (phosphorylation of the translation initiation factor eIF2), and Chop (the downstream protein of ER stress) by Western blotting analysis. Compared with controls, the protein levels of the GRP78, Atf6, Chop, and p-eIF2α in Mic19 LKO mouse liver were drastically increased (Fig. 4k, l). Moreover, the ER stress inhibitor TUDCA (tauroursodeoxycholate) significantly inhibited Mic19 LKO-induced upregulation of GRP78, Atf6, Chop, and p-eIF2α in mice (Fig. S4l and S4m). These results demonstrate that Mic19 LKO triggers the ER stress in the mouse liver.

Taken together, Mic19 LKO-caused reduction of ER-mitochondria contacts is highly related to UPRmt and ER stress.

## Mic19 LKO impairs liver fatty acid metabolism in mice

To further investigate the physiological function of Mic19-mediated ER-mitochondria contacts and the disorder of mitochondrial lipid metabolism, we investigated the effect of Mic19 LKO on lipid metabolism in mice. Mic19 LKO mice showed a significant decrease in body weight than littermate control mice (Mic19flox/flox) under normal chow

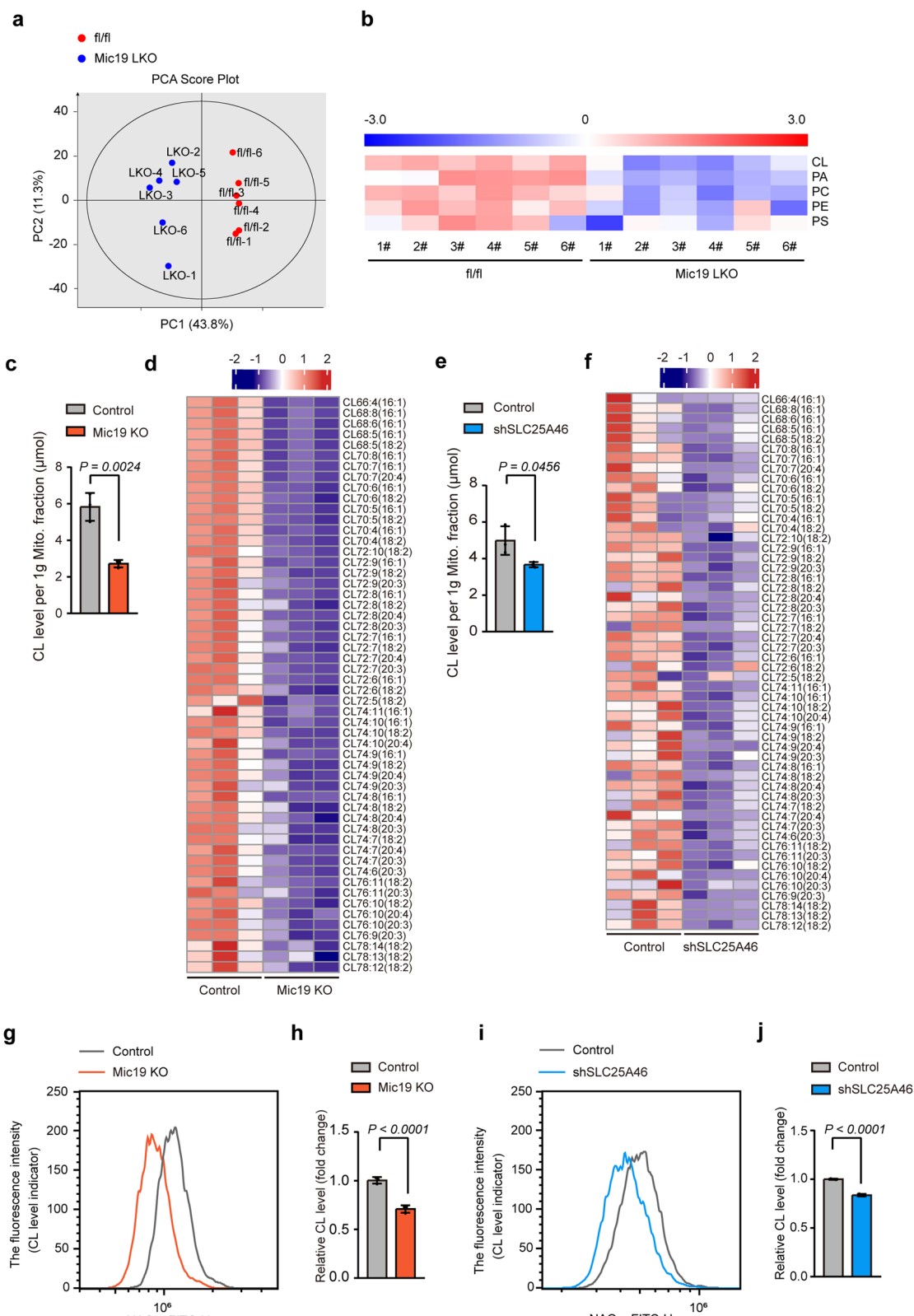

diet (Fig. S5a and S5b), which is not due to less food and water intake since Mic19 LKO mice had a significantly increased food and water intake (Fig. S5c and S5d). In addition, the level of hepatic triglyceride (TG) was significantly increased, but the level of muscle TG, serum TG, hepatic cholesterol and serum very-low-density lipoprotein (VLDL) were not changed in Mic19 LKO mice at 3 months old (Fig. 5a–e). Moreover, the levels of liver and serum free fatty acid (FFA) were dramatically elevated in Mic19 LKO mice (Fig. 5f, g). However, malonyl-CoA level and mRNA levels of most lipogenesis-related genes including *Mlycd*, *Fasn*, *Gpat2*, *Agpat1*, *Lpin1* and *Dgat2* were not changed in Mic19 LKO mouse liver (Fig. S5e and S5f). Furthermore, Western blotting analysis showed that p-Acc1/Acc1 (the ratio) and Fasn, the major enzymes responsible for de novo fatty acid synthesis, were not changed in Mic19 mouse liver (Fig. S5g and S5h), indicating that Mic19

**Fig. 3 | Abnormal crosstalk between mitochondria and ER leads to decreased cardiolipin in mitochondria. a, b** Phospholipids analysis of hepatic mitochondrial fractions using lipidomics from livers of control (Mic19[flox/flox]) and Mic19 LKO (liver-specific Mic19 knockout) mice at 3 months old (*n* = 6 mice per group). The principal components analysis (PCA) was shown (**a**). Z score values of phospholipids from different mice were analyzed and significance were exhibited by a heat map (**b**). When the levels of metabolites were lower than the average level of the whole samples, the Z-score in this sample was shown as negative. In order to show the relative abundance of phospholipids, the heatmap was shown as blue (negative Z-score) and red (positive Z-score) color. **c–f** Cardiolipin (CL) analysis of mitochondrial fractions using lipidomics from control, Mic19 KO and shSLC25A46 cells. The total cardiolipin content was shown (**c, e**). Z score values of cardiolipin from control, Mic19 KO, or shSLC25A46 cells were analyzed and significance were exhibited by a heat map (**d, f**). When the levels of CL were lower than average level of the

whole samples, the Z-score in this sample was shown as negative. In order to show the relative abundance of CL, the heatmap was shown as blue (negative Z-score) and red (positive Z-score) color. Results were representative of 3 independent experiments. Data are presented as mean ± SD, two-tailed Student's t-test. Adjustments were made for multiple comparisons. **g, h** Control and Mic19 KO HeLa cells were stained with NAO, followed with flow cytometric analysis to measure the level of Cardiolipin (CL) (**g**). 3 independent experiments were performed. Data with error bars are shown as mean ± SD (**h**). Statistical significance was assessed by two-tailed Student's t-test. **i, j** HeLa cells were infected with control (scrambled shRNA) or shSLC25A46 lentiviral particles. 5 days later, cells were stained with NAO, and CL level was measured by flow cytometric analysis (**i**). 3 independent experiments were performed. Data are presented as mean ± SD, statistical significance was assessed by two-tailed Student's t-test. *P* values are indicated in the figure. Source data are provided as a Source Data file.

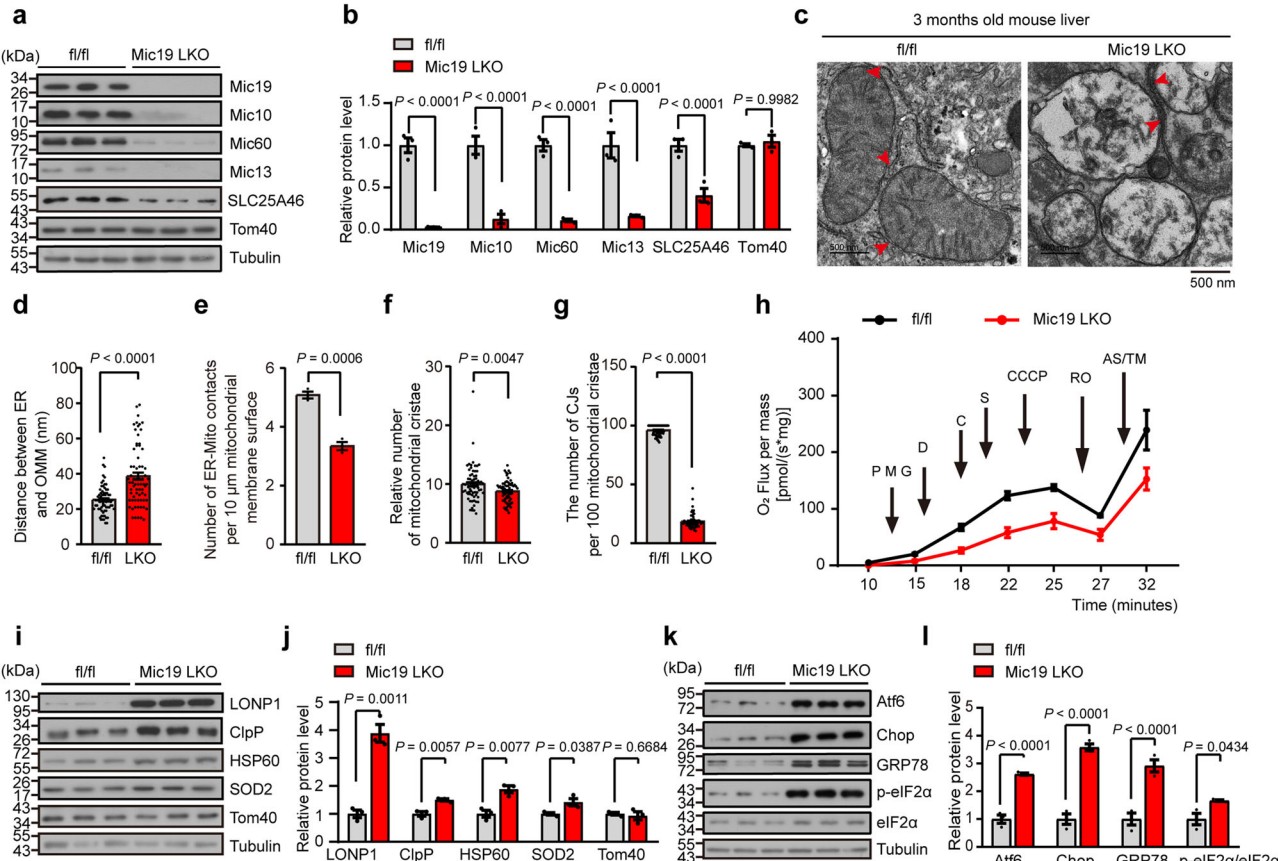

**Fig. 4 | Mic19 LKO leads to UPR^mt and ER stress in mice. a, b** Western blotting analysis of livers extracts isolated from 3-month-old Mic19[flox/flox] and Mic19 LKO mice using the indicated antibodies (**a**). Relative protein levels (the indicated protein level/Tubulin level) were further evaluated by densitometry analysis using ImageJ software (**b**). *n* = 3 mice examined over 3 independent experiments. Data are presented as mean ± SEM, statistical significance was assessed by two-way ANOVA. **c–g** Hepatic mitochondria of controls and Mic19 LKO mice aged at 3 months were analyzed by transmission electron microscopic (TEM), and the representative images were displayed (**c**). The distance between ER and outer mitochondrial membrane (OMM) (**d,** *n* = 70), the number of ER-Mito contacts per 10 μm Mitochondrial membrane surface (**e,** *n* = 3), the number of mitochondrial cristae (**f,** *n* = 70), and the number of mitochondrial cristae junctions (CJs) per 100 mitochondrial cristae (**g,** *n* = 70) were quantified and analyzed, respectively. Data are shown as mean ± SEM; statistical analysis was performed by two-tailed Student's

t-test. **h** Fresh Mic19[flox/flox] and Mic19 LKO mice liver tissues (4 mg) were homogenized in MiR05 buffer. The oxygen consumption was measured with Oroboros O2k system with the sequential addition of substrates, uncoupled, and inhibitors. *n* = 3 mice examined over 3 independent experiments. The data are presented as mean ± SEM. Substrates are as follows: CI (PMG, pyruvate + malate + glutamate), D (i.e., ADP), CII (S, succinate), and CIV (AS/TM, ascorbate + TMPD). The uncoupler is CCCP (U). Inhibitors are as follows: CI (RO, rotenone). **i–l** Livers extracts isolated from 3-month-old Mic19[flox/flox] and Mic19 LKO mice were analyzed by Western blotting analysis using the indicated antibodies (**i, k**). Relative protein levels (the indicated protein level/Tubulin level) were further evaluated by densitometry analysis using ImageJ software (**j, l**). *n* = 3 mice examined over 3 independent experiments. Data are presented as mean ± SEM, statistical significance was assessed by two-way ANOVA. *P* values are indicated in the figure. Source data are provided as a Source Data file.

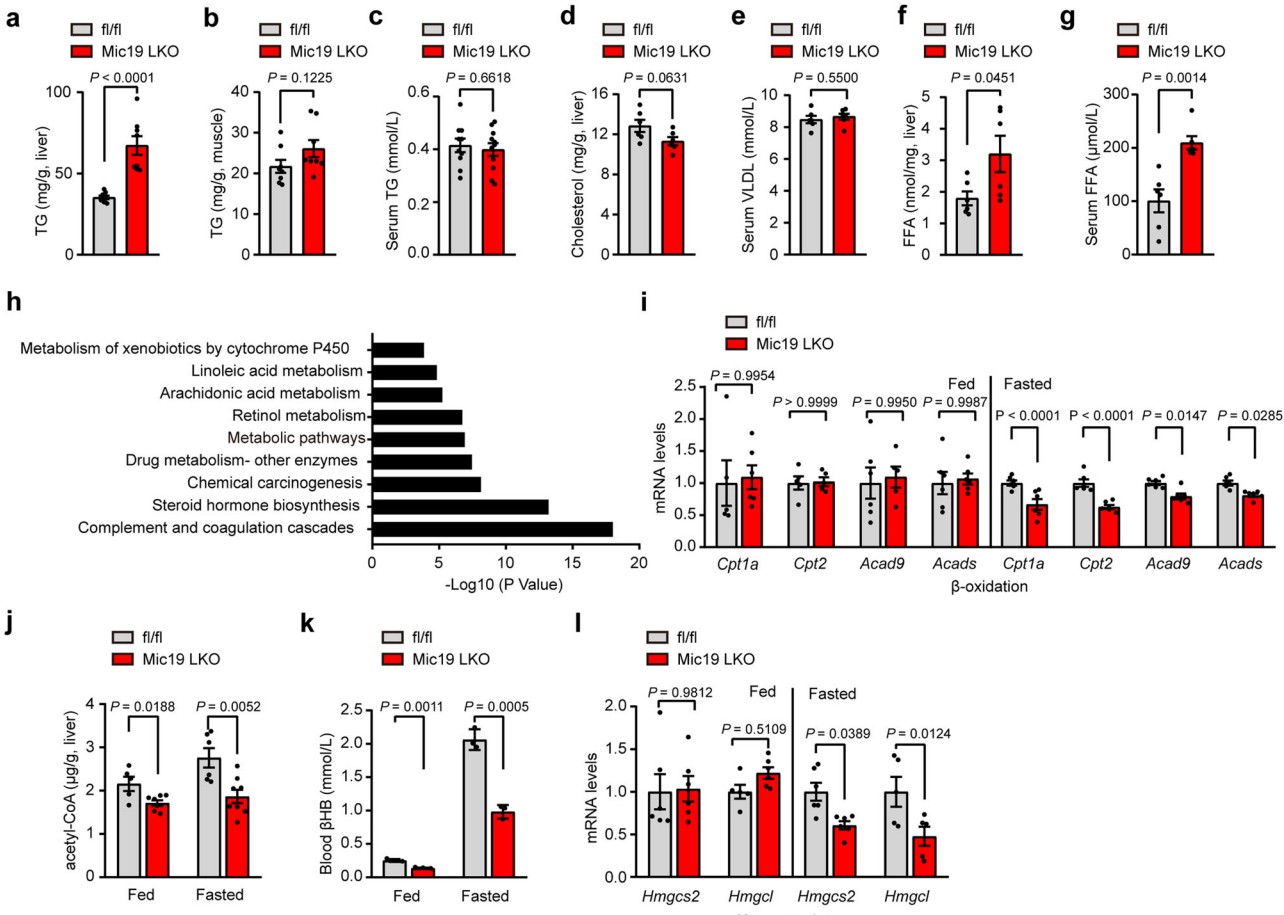

**Fig. 5 | Mic19 LKO affects fatty acid metabolism in mice. a−c** TG levels in liver (**a**), skeletal muscle (**b**), serum (**c**) of Mic19^flox/flox (n = 8 mice) or Mic19 LKO mice (n = 11 mice) aged at 3 months were detected. Data are expressed as mean ± SEM obtained from 3 independent experiments. Two-tailed Student's t-test was conducted for statistical analysis. **d−g** Cholesterol level in livers (**d**), very low-density lipoprotein (VLDL) level in serums (**e**), free fatty acid (FFA) level in serum (**g**) and liver (**f**) of Mic19^flox/flox (control) or Mic19 LKO mice aged at 3 months were detected. (n = 6 mice per group). Data are expressed as mean ± SEM obtained from at least three independent experiments. Statistical significance was further assessed by two-tailed Student's t-test. **h** Global gene expression by RNA-seq of Mic19^flox/flox (control) or Mic19 LKO mice at 3 months old (n = 3 mice per group). Gene ontology (GO) and KEGG pathway enrichment analysis of the downregulated genes compared to control mice obtained from RNA-seq results were performed. Statistical

significance was evaluated by Fisher's exact test. Adjustments were made for multiple comparisons. **i, l** Relative mRNA level of fatty acid β-oxidation related genes (*Cpt1a, Cpt2, Acad9, Acads*) (**i**) and genes (*Hmgcs2, Hmgcl*) involved in keto-genesis (**l**) in livers isolated from fed and 24-h-fasted Mic19^flox/flox (control) or Mic19 LKO mice (n = 6 mice per group) were measured by quantitative RT-PCR analysis. The mRNA levels of target genes were normalized to that of GAPDH. Data are presented as mean ± SEM; statistical analysis was assessed using two-way ANOVA. **j, k** Acetyl-CoA level in livers (**j**, n = 5 in control mice or n = 8 in Mic19 LKO mice) and blood β-hydroxybutyric acid (βHB) levels (**k**, n = 3 mice) from fed and 24-h-fasted Mic19^flox/flox (control) or Mic19 LKO mice aged at 3 months were detected. Data are shown as mean ± SEM. Statistical significance was evaluated by two-way ANOVA. P values are indicated in the figure. Source data are provided as a Source Data file.

LKO may do not impair fatty acid synthesis. These data suggest that Mic19 LKO impairs liver fatty acid metabolism but not synthesis in mice. To further explore the underlying mechanism, we extracted control and Mic19 LKO mouse liver RNA for sequencing. We found 3145 upregulated and 3133 downregulated genes in Mic19 LKO mouse liver (Fig. S5i). Interestingly, the genes related to lipid metabolism, especially fatty acid-related metabolism (metabolism of cytochrome P450, linoleic acid and arachidonic acid), were downregulated in Mic19 LKO mouse liver (Fig. 5h). Thus, we analyzed mitochondrial fatty acid oxidation of mouse liver. Fatty acids can be broken down to produce cellular energy through β-oxidation in mitochondria, and the acetyl-coA molecules produced can be eventually converted into ATP via the citric acid cycle[42]. qRT-PCR analysis revealed that the mRNA levels of some mitochondrial β-oxidation genes including *Cpt1a, Cpt2, Acad9* and *Acads* were significantly decreased in Mic19 LKO mouse liver under fasted conditions (Fig. 5i). However, ER stress inhibitor TUDCA inhibited the Mic19 LKO-induced reduction of these genes related to mitochondrial β-oxidation (Fig. S5j). Also, the level of the acetyl-coA

was significantly reduced in Mic19 LKO mouse liver under both fed and fasted conditions (Fig. 5j). These data suggest that Mic19 LKO impairs mitochondrial fatty acid β-oxidation in mouse liver.

In addition, it is well known that mitochondrial fatty acid β-oxidation can lead to ketone body production (ketogenesis) by the liver under fasted conditions[43]. We found that the ketogenesis product of β-hydroxybutyrate (βHB, one of ketone bodies) was remarkably reduced in Mic19 LKO mice (Fig. 5k), and the mRNA levels of keto-genesis related genes *Hmgcs2* and *Hmgcl* were decreased in Mic19 LKO mouse liver (Fig. 5l), indicating that ketogenesis process is impaired in Mic19 LKO mouse liver.

Also, Mic19 LKO mice displayed remarkably increased oxygen consumption (VO2) and carbon dioxide production (VCO2) rates by mice metabolism cage experiment (Fig. S5K-S5n). As adipose tissue is an important organ in regulating energy balance, serving not only as an energy store but also as a modulator of metabolism. The weight of adipose tissues including eWAT (epididymal white adipose tissue) and iWAT (inguinal white adipose tissue) were slightly decreased (Fig. S5o).

However, qRT-PCR analysis revealed that the mRNA levels of some mitochondrial β-oxidation genes including *Cpt1a, Cpt2* and *Acads* were significantly increased in Mic19 LKO mouse iWAT (Fig. S5p). These results indicate that lipolysis is increased in the iWAT of Mic19 LKO mice, which may be a compensatory response of the impairment of β-oxidation in the liver tissue, thus increasing the energy expenditure in Mic19 mice.

Thus, Mic19 LKO increases the levels of fatty acids and TG in mouse liver, which is most likely due to the reduction of hepatic mitochondrial β-oxidation.

## Mic19 LKO causes nonalcoholic steatohepatitis (NASH) and liver fibrosis

Since Mic19 LKO impairs mitochondrial β-oxidation in mouse liver, we then assessed the pathological function of Mic19 LKO in mice. Compared with control, Mic19 LKO dramatically increased the activities of alanine aminotransferase (ALT) and aspartate aminotransferase (AST) in mice at 3 months-old (Fig. 6a, b), indicating that Mic19 LKO causes liver injury in mice. Moreover, Mic19 LKO mouse (3 months old) liver displayed substantial fat accumulation (Fig. 6c). In addition, hematoxylin and eosin (H&E) staining and Oil red O staining revealed that Mic19 LKO mice (3 months old) displayed remarkably significant hepatic steatosis (Fig. 6d, e). Further TEM analysis revealed that compared to control mice, Mic19 LKO mouse (3 months old) liver showed significantly increased the number and size of lipid drops (LDs) (Fig. 6f–h). Additionally, the mRNA level of CD36 (fatty acid transporter, a marker of fatty liver disease) of Mic19 LKO mouse liver (3 months old) was more than 10 times that of control mouse liver under fed or fasted conditions (Fig. S6a), indicating that lipid uptake may contribute to Mic19 LKO-induced lipid accumulation in mouse liver. These data suggest that Mic19 LKO causes fatty liver in mice at 3 months old. Moreover, we investigated whether the Mic19 LKO-induced fat accumulation leads to liver inflammation. We detected the expression of genes encoding inflammatory cytokines and chemokines. qRT-PCR analysis revealed that the mRNA levels of inflammation-related genes *Cxcl10, Cd68, and Tnf* were significantly elevated in Mic19 LKO mouse (3 months old) livers (Fig. S6b). Immunochemical analysis CD68 staining further displayed that monocyte/macrophages are remarkably increased in Mic19 LKO mouse (3 months old) liver (Fig. S6c and S6d), confirmed that Mic19 LKO causes chronic hepatic inflammation in mice at 3 months old. Additionally, Masson's trichrome staining revealed that there was no difference between controls and Mic19 LKO mice (3 months old) (Fig. S6e and S6f). Therefore, our data suggest that Mic19 LKO leads to nonalcoholic steatohepatitis (NASH) in mice (3 months old).

Chronic liver inflammation can lead to liver fibrosis[44]. We next investigated whether Mic19 LKO leads to liver fibrosis as a consequence of the natural progression of liver disease. Mic19 LKO mice (7 months old) still showed a lower body weight than littermate controls under normal chow diet (Fig. S7a). Additionally, the serum levels of ALT and AST were still significantly increased in Mic19 LKO mice at 7 months old (Fig. 6i, j). However, compared to control, Mic19 LKO did not show remarkable fat accumulation in mice at 7 months old (Fig. 6k). In addition, hepatic TG levels did not differ between control and Mic19 LKO mice (7 months old) (Fig. 6l), probably due to compensatory effect in mice. While, H&E staining revealed significantly increased inflammation and necrosis areas in Mic19 LKO mouse (7 months old) liver (Figs. 6m and S7b), and qRT-PCR analysis showed that the mRNA levels of inflammation-related genes including *Cxcl10, Cd68,* and *Tnf* were remarkably increased in Mic19 LKO mouse (7 months old) livers (Fig. S7c). Moreover, Masson's trichrome staining showed that compared to control, Mic19 LKO (7 months) cause a dramatic accumulation of extracellular collagen in mouse liver (Fig. 7n and S8d), indicating that Mic19 LKO causes liver fibrosis in mice (7 months old). Also, qRT-PCR analysis showed that the mRNA levels of genes

encoding fibrotic markers including *Col1a1* and *Col3a1* were remarkably increased in Mic19 LKO mice (7 months old) (Fig. S7e). Additionally, the level of hydroxyproline (a fibrotic marker in liver tissues), serum alkaline phosphatase (ALP, a marker of liver disease) and γ-glutamyl transpeptidase (γ-GT, a marker of the liver cell injury) were also significantly increased in Mic19 LKO mice (7 months old) (Fig. S7f-S7h). These data suggest that Mic19 LKO provokes a progressive increase in liver fibrosis.

Together, Mic19 LKO progressively triggers NASH and liver fibrosis in mice.

## Mic19 re-expression in Mic19 LKO mice restores liver lipid metabolism and blocks liver diseases

To determine whether the alterations in Mic19 LKO mice were attributable to Mic19 loss-of-function mechanisms, we injected tail intravenous administration of adeno-associated virus (AAV) encoding control or Mic19-Flag into 8-week-old Mic19 LKO mice to re-expressed Mic19 in mouse liver. After Mic19 re-expression in Mic19 LKO mouse liver (Fig. 7a), the body weight of mice recovered to that of control (Mic19$^{flox/flox}$) mice (Fig. S8a and S8b), and the protein levels of UPR$^{mt}$-related proteins including LONP1, ClpP, HSP60 and SOD2 were significantly lower than those of Mic19 LKO mouse liver, and there was no difference between them and that of the control mouse liver (Fig. 7b, c), indicating that Mic19 re-expression dramatically inhibits Mic19 LKO-induced UPR$^{mt}$. In addition, we examined the impact of Mic19 re-expression on ER stress in mice. Western blotting showed that the protein levels of the GRP78, Atf6, Chop and p-eIF2α in Mic19 re-expressed Mic19 LKO mouse livers were significantly lower than those of Mic19 LKO mouse livers, but were similar to those of control mouse livers (Fig. S8c-S8f), suggesting that Mic19 re-expression remarkably attenuates Mic19 LKO-caused ER stress in mice.

Then, we investigated the effect of Mic19 re-expression on mouse liver lipid metabolism. Upon Mic19 re-expression in Mic19 LKO mouse liver, the level of hepatic triglyceride (TG) was significantly lower than that of Mic19 LKO mice (Fig. 7d). Consistently, Oil red O staining showed that Mic19 re-expression dramatically inhibited Mic19 LKO-caused hepatic steatosis (Fig. 7e). In addition, compared to Mic19 LKO, Mic19 re-expression remarkably downregulated the mRNA levels of liver inflammation-related genes including *Cxcl10, Cd68,* and *Tnf* (Fig. 7f). Moreover, Mic19 re-expression remarkably decreased the serum level of ALT and AST in mice (Fig. 7g, h). These data suggest that Mic19 re-expression blocks Mic19 LKO-trigged NASH by restoring liver lipid metabolism.

We also investigated whether Mic19 re-expression inhibits Mic19 LKO-caused liver fibrosis in mice (7 months old). H&E and Masson's trichrome staining displayed that Mic19 re-expression significantly decreased Mic19 LKO (7 months)-caused necrosis and accumulation of extracellular collagen in mouse liver (Fig. 7i–l). Additionally, qRT-PCR analysis showed that the mRNA levels of genes encoding fibrotic markers *Col1a1* and *Col3a1* in Mic19 re-expressed Mic19 LKO mouse (7 months old) liver were remarkably lower than those in Mic19 LKO mouse liver (Fig. 7m). These data suggest that Mic19 re-expression blocks Mic19 LKO (7 months)-trigged liver fibrosis in mice.

Therefore, Mic19 re-expression in Mic19 LKO mice blocks Mic19 LKO-trigged NASH and liver fibrosis by restoring liver lipid metabolism.

## Mic19 overexpression suppresses MCD-induced fatty liver disease

To further confirm the connection between Mic19 and NASH, we established a NASH mouse model (termed MCD), which consists of a methionine and choline-deficient diet combined with 45% high fat diet (HFD) and supplemented with 0.1% L-methionine in drinking water. H&E and Oil red O staining revealed that MCD mice showed significant increased fat accumulation (Fig. 8a, b). In addition, TG

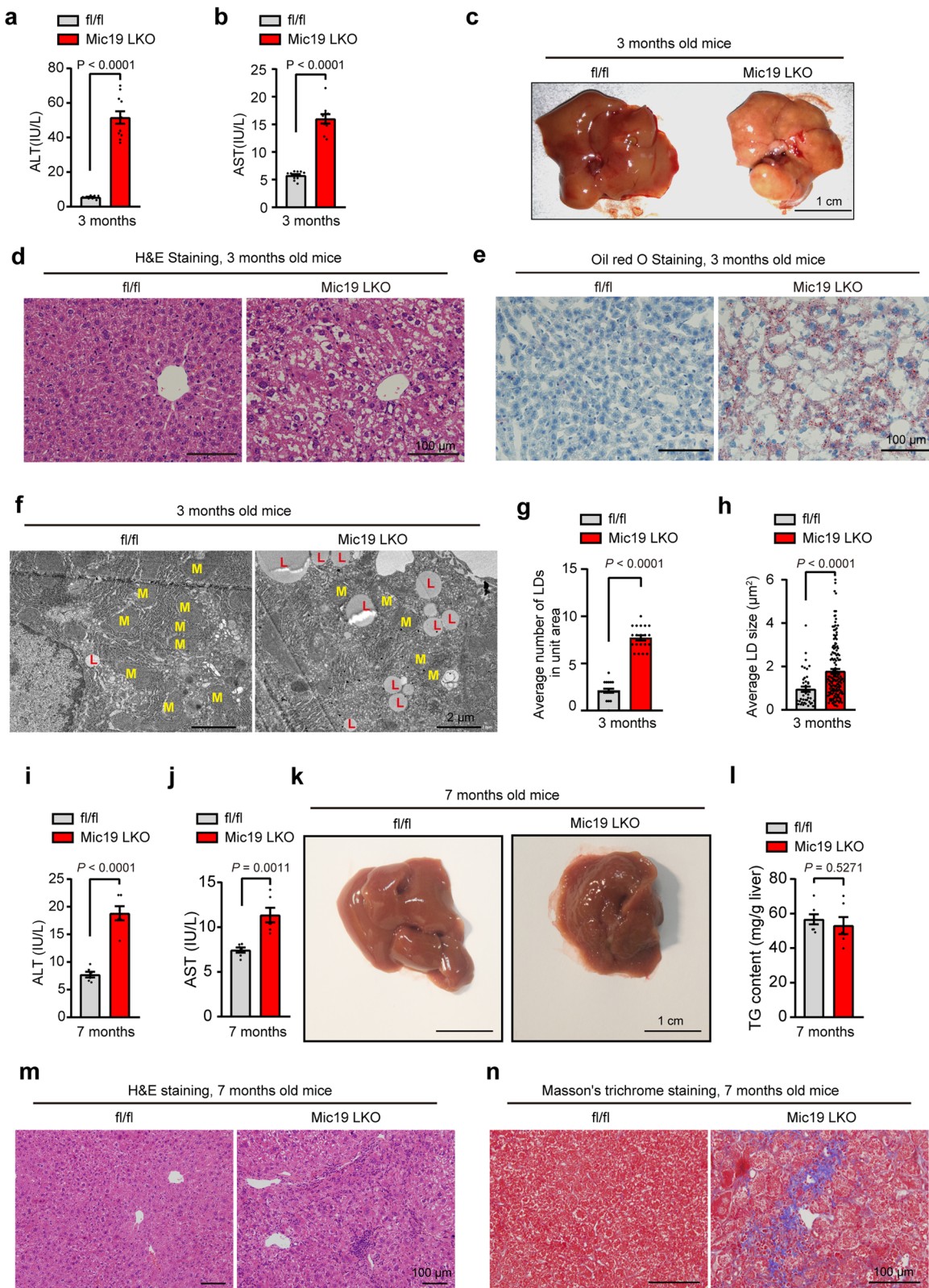

level and mRNA level of inflammation-related genes *Cd68, Cxcl10 and Tnf* were significantly increased in MCD mouse liver (Fig. 8c, d). These data suggest that MCD causes NASH (fatty liver and liver inflammation) in mice. We then examined the expression of Mic19 in control and MCD mice. Mic19 and Mic60 protein expression in mouse liver was significantly lower in the MCD mice than that in controls (Fig. 8e, f). Moreover, TEM analysis revealed that MCD mouse liver cells

showed significant decreased mitochondrial cristae and dramatic decreased number of mitochondrial cristae junctions (Fig. 8g–i), indicating that MCD induces mitochondrial cristae remodeling, consistent with the data that MCD mouse has low protein level of Mic19 in liver. Interestingly, ER-mitochondria contacts were significantly decreased in MCD mouse liver cells comparing with controls (Fig. 8g, j).

**Fig. 6 | Mic19 LKO causes NAFLD in mice. a, b** Serum alanine aminotransferase (ALT) (**a**) and aspartate aminotransferase (AST) (**b**) activities of Mic19$^{flox/flox}$ (control, n = 9) or Mic19 LKO mice (n = 11 in Mic19 LKO mice) at 3 months old were analyzed. Data are shown as the mean ± SEM, statistical significance was performed by two-tailed Student's t-test. **c** Representative images of livers from littermate 3-month-old male Mic19$^{flox/flox}$ (control) and Mic19 LKO mice. **d, e** Liver sections from Mic19$^{flox/flox}$ (control) or Mic19 LKO mice aged 3 months with a normal diet were analyzed by H&E staining (**d**) or Oil red O staining (**e**). Representative images were displayed. All data are representative of 3 independent experiments. **f–h** Mitochondrial ultrastructure in liver sections from Mic19$^{flox/flox}$ (control) or Mic19 LKO mice were analyzed by TEM analysis (**f**, n = 6 mice). The yellow "M" indicates mitochondria. The red "L" indicates lipid. The number (**g**, n = 20) and size (**h**, n = 42 in control, n = 154 in Mic19 LKO mice) of lipid droplets (LD) were calculated and analyzed by ImageJ software. Data are presented as mean ± SEM; statistical analysis was determined using two-tailed Student's t-test. **i, j** Serum ALT (**i**) and AST (**j**) activities of Mic19$^{flox/flox}$ (control) or Mic19 LKO mice (n = 6 mice per group) at 7 months old were detected. Data are shown as the mean ± SEM, statistical significance was evaluated by two-tailed Student's t-test. **k** Representative images of livers from littermate 7-month-old male Mic19$^{flox/flox}$ (control) and Mic19 LKO mice. **l** Hepatic triglyceride (TG) levels from Mic19$^{flox/flox}$ (control) or Mic19 LKO mice (n = 6 mice) aged 7 months were measured. Data are expressed as mean ± SEM and two-tailed Student's t-test was conducted for statistical analysis. **m, n** Liver tissues from Mic19$^{flox/flox}$ (control) or Mic19 LKO mice aged 7 months were fixed with formaldehyde, and the liver sections were used for H&E (**m**) and Masson's trichrome staining (**n**). All data are representative of 3 independent experiments. *P* values are indicated in the figure. Source data are provided as a Source Data file.

Then, we investigated the effect of Mic19 overexpression on MCD-induced liver disease. adeno-associated virus encoding Mic19-Flag were injected by a tail vein into 8-week-old MCD mice. After overexpression of Mic19-Flag in MCD-treated mouse liver, the upregulation of TG level and mRNA level of inflammation-related genes *Cd68, Cxcl10 and Tnf* were suppressed and recovered to be similar to that of normal diet mice (Fig. 8k–m). Moreover, H&E staining revealed that Mic19-Flag overexpression remarkably suppressed MCD-caused hepatic steatosis in mice (Fig. 8n). Therefore, Mic19 overexpression suppresses MCD-induced fatty liver disease.

These results are consistent with the function of Mic19 in repressing mouse liver disease.

## Discussion

Phospholipids are the main components of mitochondrial membranes, and the synthesis and transport of mitochondrial phospholipids are key to the maintenance of mitochondrial structure and function. Most of the mitochondrial phospholipids are synthesized in the ER, and then transported to the mitochondria through ER-mitochondria contacts, but the mechanism and physiological function of ER-mitochondria contacts and mitochondrial phospholipids metabolism remain largely unknown. Here, we show that the Mic19-SLC25A46-EMC2 axis regulates mitochondria-ER contacts. Mic19 deficiency causes the disorder of mitochondrial phospholipids metabolism including the reduction of CL production, then impairing mitochondrial membrane organization, leading to impaired fatty acid metabolism. Strikingly, Mic19 LKO results in NASH and liver fibrosis. Our study demonstrates that ER-mitochondria contacts are critical for mitochondrial membrane organization, and the impairment of ER-mitochondria contacts is involved in the development and progression of liver diseases.

ER-mitochondria contacts play an important role in a variety of cellular activities, including lipid synthesis and transport, Ca$^{2+}$ transport, ER stress, apoptosis, autophagy, and mitochondrial fission[45]. ER-mitochondria contacts have been reported to be associated with the transport of PS[19,30]. However, the molecular mechanism of ER-mitochondrial contacts remains largely obscure, and the related factors and pathways need to be further identified. Herein, we found that the EMC2-SLC25A46-Mic19 axis is involved in ER-mitochondria contacts (Fig. 2), consisting with the previously report that SLC25A46 regulates mitochondrial lipid homeostasis[35]. Interestingly, the deficiency of Mic19 or SLC25A46 reduces the mitochondrial phospholipids especially the level of CL (Fig. 3c–j, and S3d-S3g), indicating that Mic19 and SLC25A46 is involved in mitochondrial phospholipid metabolism. Additionally, Mic19 locates in mitochondrial inner membrane space[34], suggesting that Mic19 may participate in mitochondrial phospholipid transport from OMM to IMM. It has been reported that TRIAP1/PRELI complex is required for the transport of PA between OMM and IMM in mammalian cells[46]; in addition, PTPIP51 can bind to and transfer PA in vitro[47], but depletion of PTPIP51 reduces but not totally inhibits cardiolipin production in cells. These reports suggest that multiple pathways may contribute to ER-mitochondrial PA transport in mammalian cells. Therefore, the EMC2-SLC25A46-Mic19 axis may be a pathway to regulate mitochondrial phospholipid metabolism. In addition, Mic19 and SLC25A46 regulate mitochondrial cristae junctions maintenance and cristae remodeling (Figs. 1e, g, 2i, k, 5c, 5f, g), demonstrating that ER-mitochondria contacts are critical for the organization of mitochondrial membranes, especially cristae membranes. It should be noted that PC, PS, and PE were also reduced in mitochondria of Mic19 KO cells (Fig. 3b). It is probably because PC and PS also transported from ER to mitochondria, and the Mic19 KO-caused the reduction of ER-mitochondria contacts may impair the transport of PC and PS to mitochondria. In addition, PS is the precursor for PE, the decreased PE may be the consequence of the reduced PS. The ER-mitochondria contacts and mitochondrial phospholipid metabolism may contribute to the preservation of the cristae structure of mitochondria and conduce to ensure the mitochondrial related physiological processes.

MICOS complex is critical for mitochondrial membrane organization and mediates OMM-IMM contacts[21,34], and is involved in PS transport and PE synthesis[29,30]. The mutation or dysfunction of MICOS complex is assciated with a series of diseases including cardiovascular disease, Parkinson's disease, diabetes, liver disease, and cancer[21,32,33]. However, the pathologic mechanisms by which MICOS dysfunction leads to these diseases remain unclear, and mammalian disease models for MICOS are still lacking. Here, we create a Mic19 (a key subunit of MICOS complex) liver-specific knockout (LKO) mouse model. Mic19 LKO hepatic mitochondria display loss of mitochondrial cristae junctions and reduction of cristae (Figs. 4c, f, g), suggesting that Mic19 LKO impairs mitochondrial membrane organization in mice liver cells. In addition, Mic19 deletion results in a reduction of the MICOS complex subunits, including Mic60 (Fig. 4a, b), Mic26, and Mic27[24]. Mic60 could co-sediment with liposomes and regulates membrane shaping[48], and Mic26 or Mic27 are also associated with cardiolipin metabolism[49]. Moreover, the alterations of inner membrane morphology may be linked to membrane lipid metabolism and synthesis[50], thus, alterations of inner membrane morphology may potentially influence the localization of enzymes and proteins on the mitochondrial inner membrane and impair phospholipid synthesis. Therefore, Mic19 may directly or indirectly regulate lipid metabolism and transport. Interestingly, Mic19 LKO mice develop a NASH and liver fibrosis phenotype (Fig. 6), and Mic19 re-expression in Mic19 LKO mice liver remarkably blocks the progression of Mic19 LKO-induced NASH phenotype (Fig. 7). Additionally, the protein levels of Mic19 and Mic60 are significantly reduced in MCD-treated mice with NASH and Mic19 overexpression remarkably suppresses MCD-caused live diseases in mice (Fig. 8e, f, and S8g-j). These findings suggest that Mic19 (or MICOS complex) dysfunction-induced abnormal phospholipid metabolism and mitochondrial structure are highly associated with the development and progression of liver disease. Moreover, the Mic19 LKO mouse model can be of great benefit for understanding the mechanism of liver disease and assessing the therapeutic potential and effect of anti-

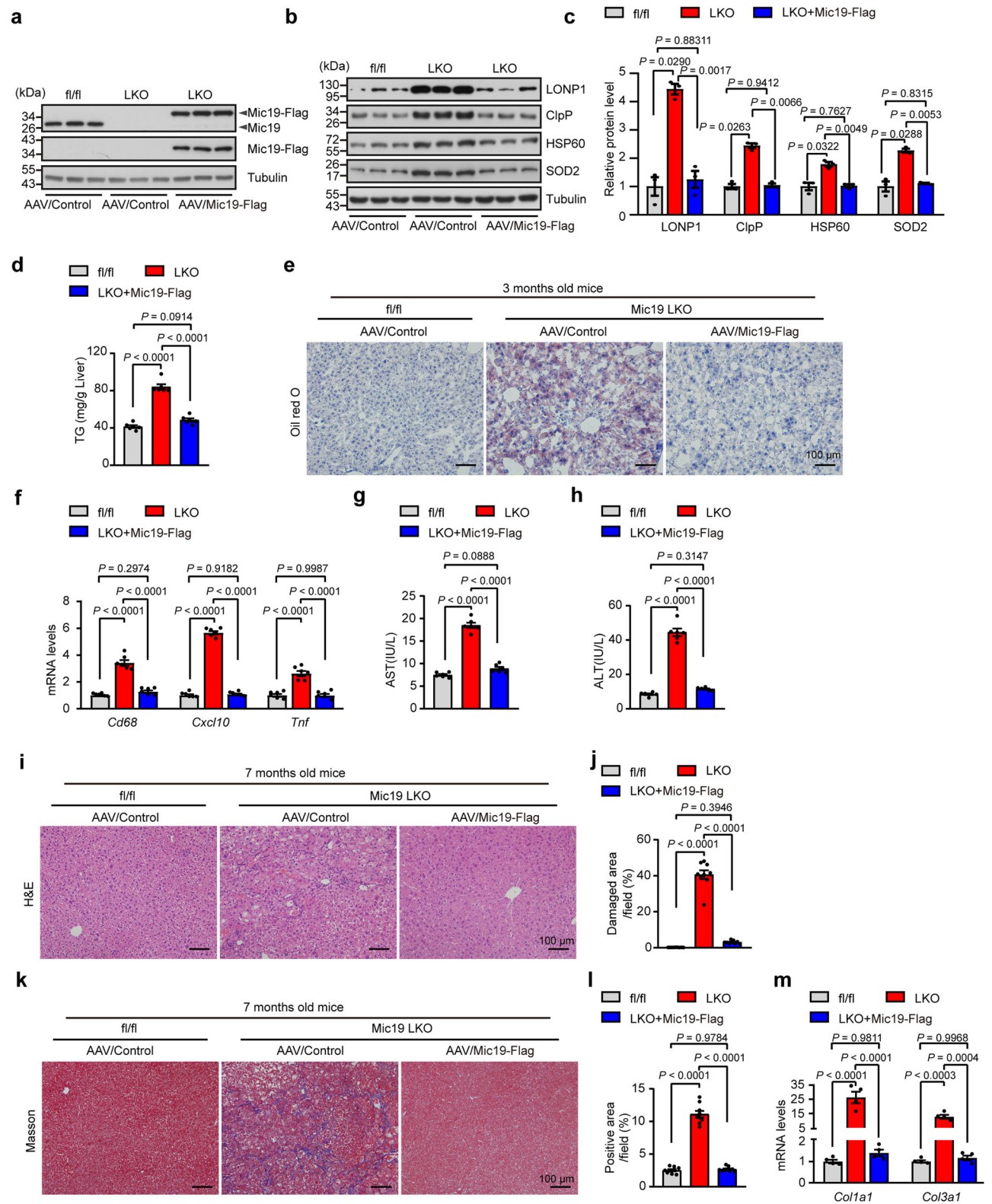

NASH or anti-liver fibrosis drug candidates. In terms of pathological mechanism, Mic19 depletion leads to the decreased ER-mitochondria contacts, impairing mitochondrial phospholipid metabolism and mitochondrial membrane disorganization (including the reduction of mitochondrial cristae and loss of mitochondrial cristae junctions) and UPR^mt, then causing mitochondrial dysfunctions such as weakened mitochondrial respiration, reduced ATP production and increased ROS levels, etc. (Figs. 1, 3, 4 and S4). As mitochondrion is a

crucial organelle for fatty acid oxidation, its dysfunction also impairs mitochondrial β-oxidation, leading to hepatic lipid accumulation, NASH and liver fibrosis in Mic19 LKO mice (Figs. 5, 6 and S9). Therefore, the impairment of ER-mitochondrial contacts and mitochondrial phospholipid metabolism may be a driving factor in the development and progression of liver disease, and MICOS complex might be a target in drug development to stop the progression of liver disease.

**Fig. 7 | Mic19 re-expression ameliorates liver injury in Mic19 LKO mice. a–c** The male 8-week-old Mic19 LKO mice were injected with $1\times10^{11}$ adeno-associated virus expressing control or Mic19-Flag via tail vein. At 3 months old, the mice were euthanized. Liver lysates were analyzed by Western blotting analysis with indicated antibodies (**a, b**). Relative protein levels to Tubulin) were evaluated by using ImageJ software (**c**). $n = 3$ mice. Data are presented as mean ± SEM, statistical significance was assessed by two-way ANOVA. **d, e** Hepatic TG levels (**d**) and Oil red O staining (**e**) were detected from Mic19$^{flox/flox}$ (control), Mic19 LKO, or Mic19 re-expression Mic19 LKO mice ($n = 6$ mice per group). Data are shown as mean ± SEM, statistical significance was assessed by one-way ANOVA. All data are representative of 3 independent experiments. **f–h** Relative liver mRNA levels of inflammatory cytokines, serum ALT (**g**), and serum AST (**h**) from 3-month-old Mic19$^{flox/flox}$ (control), Mic19 LKO, or Mic19 re-expression Mic19 LKO mice ($n = 6$ mice) to GAPDH were determined. Data are presented as mean ± SEM; statistical analysis was assessed using two-way ANOVA. **i–l** The male 8-week-old Mic19 LKO mice were injected with $1\times10^{11}$ adeno-associated virus expressing control or Mic19-Flag via tail vein. At 7 months old, the mice were euthanized. Liver damaged areas in H&E staining images (**i, j**) and Masson's trichrome staining images (**k, l**) were analyzed and quantified by ImageJ software. $n = 9$ fields examined over 3 independent experiments. Results shown were representative of 3 independent experiments. Data are shown as mean ± SEM, statistical significance was assessed by one-way ANOVA. **m** Quantitative RT-PCR analysis of collagen related genes (*Col1a1, Col3a1*) in liver tissues from 7-month-old Mic19$^{flox/flox}$ (control), Mic19 LKO, or Mic19 re-expression Mic19 LKO mice ($n = 4$ mice per group). Data are presented as the mean ± SEM, statistical significance was evaluated by two-way ANOVA. *P* values are indicated in the figure. Source data are provided as a Source Data file.

Thus, our findings reveal that the EMC2-SLC25A46-Mic19 axis regulates ER-mitochondrial contacts and mitochondrial phospholipid metabolism, and highlight that ER-mitochondrial contacts are critical for the organization of mitochondrial membranes and physiologically significant for the development of liver diseases.

## Methods

### Study approval

All animal experiments were performed according to the guidelines of the China Animal Welfare Legislation and Use Committee of Wuhan University.

### Animal model and cell culture

C57BL/6 mice were hosted under specific-pathogen-free conditions at $22 \pm 2\,°C$ with a humidity of $40 \pm 5\%$ in single ventilated cages in a 12 h/12 h day/night cycle and fed with unlimited access to food and water. Mice were fed with regular chow (#1025, Beijing Huafukang, Beijing, China). For MCD experiment, 8-week-old C57BL/6 male mice were fed a standard of methionine choline-deficient diet. After 5 weeks of treatment, the mice were sacrificed and livers were surgically removed for experiments. Mic19 floxed mice were generated by conventional gene targeting of mouse embryonic stem cells (ESC) derived from C57BL/6 mice using the Mic19 gene targeting constructs designed to insert intronic LoxP sites flanking exon 2 of *Mic19* gene. For generation of Mic19 liver specific knockout (LKO) mice, Mic19 floxed mice were crossed with Alb-Cre transgenic mice.

Human cervical cancer HeLa cell line (ATCC), Monkey kidney COS-7 cell line (ATCC), and Human embryonic kidney 293 T cells (ATCC) were cultured in DMEM (Gibco, 10566016) in supplemented with 10% FBS (Gibco, 10091155), 50 U/mL penicillin/streptomycin (Gibco, 15140122) at $37\,°C$ in an incubator equipped with 5% $CO_2$.

### Western blotting and co-immunoprecipitation

Western blotting analyses were performed as previously described[51]. Briefly, both mouse liver and cell samples were lysed by using RIPA buffer (150 mM sodium chloride, 1% Triton X-100, 0.5% sodium deoxycholate, 0.1% sodium dodecyl sulfate, 50 mM Tris, pH 8.0) with inhibitors cocktail (Roche). Subsequently, tissue or cell samples were boiled for 15 min with equal volume of 2× SDS loading buffer. Then 30 μg of proteins per lane were fractionated in the SDS-PAGE, and blotted onto PVDF membranes (Merck Millipore). The membranes were blocked at room temperature with 5% no-fat milk in TBST solution (Tris-base buffer, 0.1% Tween-20), followed by immunoblotted with the indicated primary antibodies and corresponding secondary antibodies-conjugated HRP. Finally, the PVDF membranes were visualized by using ECL chemiluminescence reagent (BIO-RAD, 1705060).

For co-immunoprecipitation (co-IP) analysis, cells were harvested and lysed in lysis buffer (150 mM NaCl, 10% glycerol, 20 mM Tris–HCl pH = 7.4, 2 mM EDTA, 0.5% NP-40, 0.5% Triton X-100 and protease inhibitor mixture) for 1 h on ice. Then lysates were centrifuged at $4\,°C$ for 15 min with $12,000 \times g$. Subsequently, supernatants were incubated with anti-FLAG M2 affinity gel (Sigma-Aldrich, A2220) or control IgG (Thermo Fisher, 10004D) for overnight at $4\,°C$ in a shaker. Next day, agarose beads were rinsed 6 times with IP lysis buffer. Then 1× PBS and equal volume of 2× SDS loading buffer were added to the agarose beads, and the samples were boiled for 8 min at $105\,°C$ in metal bath. Samples were analyzed by SDS-PAGE and Western blotting.

### Immunostaining and confocal imaging

Briefly, cells stably expressing mito-DsRed (a mitochondrial marker, red) or/and Sec61β-GFP (an ER marker, green) were infected with relevant lentiviral particles. Subsequently, cells were incubated with NAO for 30 min, and washed three times with PBS. Cells were then analyzed using a Zeiss LSM 880 confocal microscope with Airyscan. For the super-resolution imaging with HIS-SIM (High Sensitivity Structured Illumination) analysis, cells stably expressing mito-DsRed (a mitochondrial marker, red) and Sec61β-GFP (an ER marker, green) were merely infected with relevant lentiviral particles. HIS-SIM, a microscope developed from Hessian structured illumination microscopy (Hessian-SIM), is provided by the Guangzhou Computational Super-resolution Biotech Co., Ltd.

For mouse liver tissues, the sections were firstly de-paraffinized in xylene, followed by rehydrated with a gradient of ethanol from 100% to 50%. Sections were washed in distilled water. Then, the rest of the procedure is the same as that for the cells. NIH Image J was employed to analyze images.

### Transmission electron microscopy

The procedure for transmission electron microscopy (TEM) was performed according to the previous study[52]. Mice were anesthetized with intraperitoneal injection of sodium pentobarbital (50 mg/kg). Mice were fixed by cardiac perfusion of 4% paraformaldehyde in PBS for 10 minutes at room temperature. After perfusion, liver samples were dissected, cut into small pieces, and post-fixed in the 2% glutaraldehyde in the PBS (pH 7.4) overnight at $4\,°C$. Then picking up the specimens, were washed with PBS (pH 7.4). Then, tissue samples were fixed 2% $OsO_4$ for 1.5 h at $4\,°C$. Followed by washes, samples were gradually dehydrated in ethanol with increasing concentrations ranging from 30 to 100%. After dehydration, specimens were embedded in EPON resin. Finally, 70 nm thickness sections were obtained, followed by stained with 2% uranyl acetate and lead citrate. For cultured cells, Cells were first fixed with 1.5% glutaraldehyde and 3% PFA in 0.1 M cacodylate, 0.05% $CaCl_2$ buffer for 1 h at room temperature. After 3 washes with 0.1 M cacodylate buffer, the cells were fixed with 1% $OsO_4$ on ice for 1 h. Then, osmic acid was discarded and the cells were stained with 2% uranyl acetate overnight at room temperature. After several washes, cells were gradually dehydrated in ethanol with increasing concentrations ranging from 30 to 100%. The following steps are similar to that for tissue samples. Images were acquired by a transmission electron microscope (Joel Ltd, Tokyo, Japan). Image J software was used to analyze the structure of mitochondria. Additionally, both the acquisition of electron microscope images was

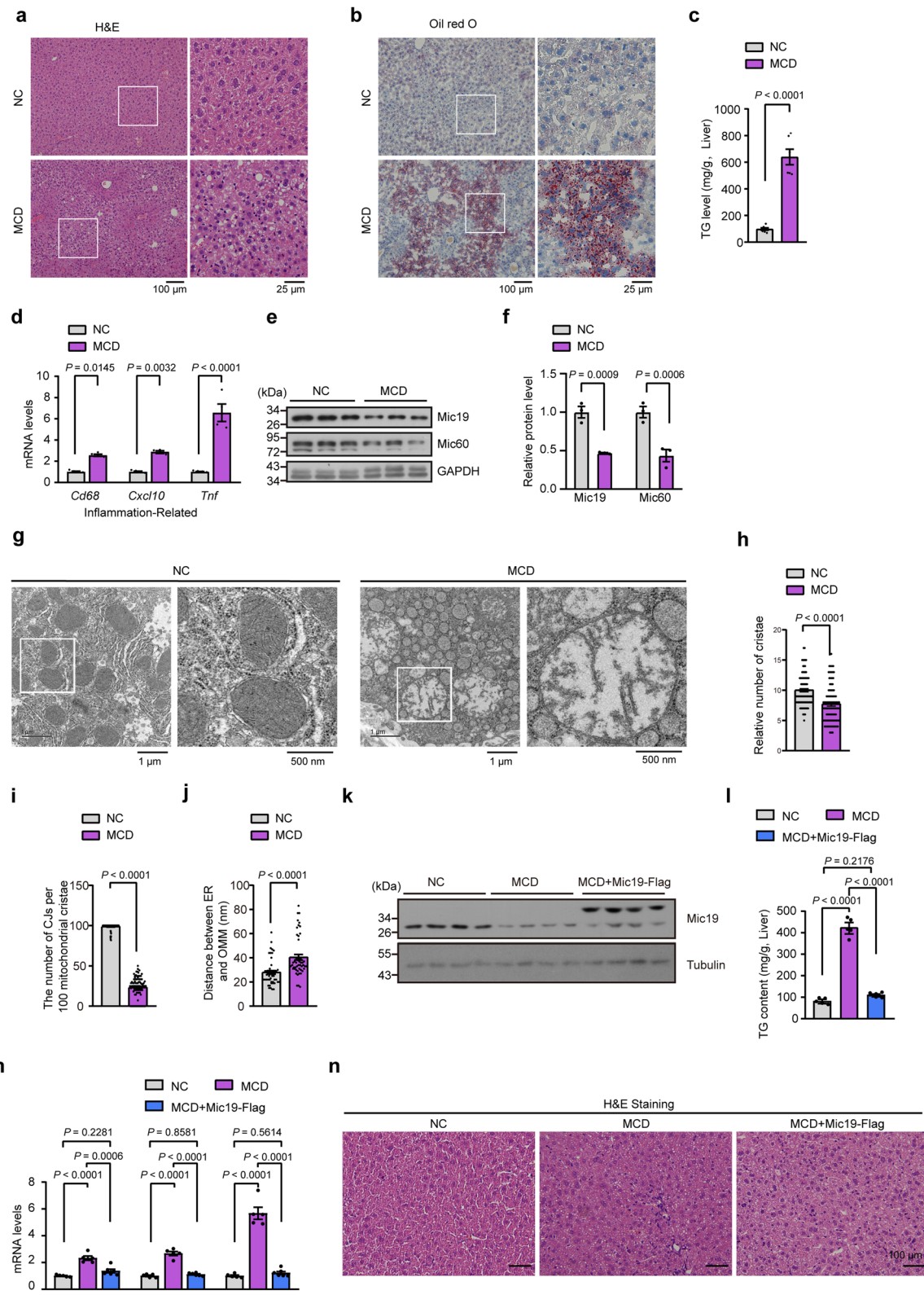

randomly performed and the statistical analysis was performed in a double-blind fashion

**Statistics and reproducibility**

Data are presented as mean ± SEM unless specified. Statistical significances between the two groups were determined using two-tailed Student's t-tests. Multiple groups were analyzed using two-way ANOVA followed. *P* values are indicated in the figure. All samples/animals were randomly grouped. All data were analyzed using GraphPad Prism software 8.0. For most in vitro experiments, the results shown in the paper present the combined results of three independently repeated experiments.

**Fig. 8 | Mouse models with MCD-induced fatty liver disease show low hepatic levels of Mic19. a, b** The male2-month-old C57BL/6 mice were fed with normal (control) or a methionine and choline-deficient (MCD) diet for 5 weeks. The H&E (**a**) and Oil red O staining (**b**) were shown. Results were representative of 3 independent experiments. **c** Hepatic TG levels were detected from normal (control, $n = 6$ mice) or MCD mice ($n = 7$ mice). Data are shown as mean ± SEM, statistical significance was assessed by two-tailed Student's t-test. **d–f** Relative mRNA level of inflammatory genes (**d**, $n = 4$ mice per group) and the protein level of Mic19, Mic60 (**f**, $n = 3$ mice per group) in normal (control) and MCD mice to GAPDH were measured by quantitative RT-PCR analysis or Western blotting analysis. Data are presented as mean ± SEM. Statistical significance was further calculated by two-way ANOVA. **g–j** Mitochondrial ultrastructure images in the liver of normal (control) or MCD mice were displayed by TEM (**g**). The number of mitochondrial cristae

(**h**, $n = 90$ mitochondria), the number of mitochondrial cristae junctions (CJs) per 100 mitochondrial cristae (**i**, $n = 90$ mitochondrial cristae), and the distance between ER and OMM (**j**, $n = 51$ mitochondria) were analyzed by ImageJ software. Data are presented as mean ± SEM, and $P$ value was measured using two-tailed Student's t-test. **k** The 8-week-old C57BL/6 male mice were injected with a control or with adeno-associated viruses encoding Mic19-Flag and then were fed with MCD for 5 weeks. The liver lysates were analyzed by Western blotting with antibodies against Mic19, Tubulin ($n = 4$ mice per group). **l–n** Hepatic TG levels (**l**), relative mRNA level of inflammatory genes to GAPDH (**m**), and H&E staining (**n**) were detected from normal (NC), MCD, or Mic19 overexpression mice ($n = 6$ mice per group). Data are shown as mean ± SEM, statistical significance was assessed by one-way ANOVA or two-way ANOVA. $P$ values are indicated in the figure. Source data are provided as a Source Data file.

## Reporting summary

Further information on research design is available in the Nature Portfolio Reporting Summary linked to this article.

## Data availability

The RNA-seq datasets in the paper are available in BioProject database under accession code PRJNA940342 and BioSample database under accession code SAMN33562657. Source data are provided in this paper.

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

## Acknowledgements

This work is supported by the Ministry of Science and Technology of the People's Republic of China (2018YFA0800700), the National Natural Science Foundation of China (32125011, 31970711, and 91854107), and supported by "the Fundamental Research Funds for the Central Universities (2042022dx0003)".

## Author contributions

Z.S. and H.H. designed the project. J.D. and L.C. performed the most experiments and data analyses. F.Y., J.T., Bing Liu, and J.L. helped complete part of the experiments. J.D., L.C., F.Y., Z.S., and H.H. wrote the manuscript. P.-H.Z., Bin Lu, M.W., J.-H.L., J.-J.H., S.E., and Q.M. edited the manuscript and provided guidance on experimental design and interpretation. All authors discussed and interpreted the data together.

## Competing interests

The authors declare no conflict of interest.

## Additional information

[1]College of Life Sciences, TaiKang Center for Life and Medical Sciences, Frontier Science Center for Immunology and Metabolism, Department of Anesthesiology, Renmin Hospital of Wuhan University, Wuhan University, Wuhan, Hubei, China. [2]Department of pathology, School of Basic Medicine, Tongji Medical College and State Key Laboratory for Diagnosis and Treatment of Severe Zoonotic Infectious Diseases, Huazhong University of Science and Technology, Wuhan, Hubei, China. [3]Department of Biochemistry and Molecular Biology, School of Basic Medical Sciences, Hengyang Medical School, University of South China, Hengyang, Hunan, China. [4]State Key Laboratory of Quality Research in Chinese Medicine, Institute of Chinese Medical Sciences, University of Macau, Macau, China. [5]Department of Pediatric Intensive Care Unit, Anhui Provincial Children's Hospital, Hefei, Anhui, China. [6]Department of Biochemistry, Rappaport Faculty of Medicine, Technion-Israel Institute of Technology, Haifa, Israel. [7]These authors contributed equally: Jun Dong, Li Chen, Fei Ye. ✉e-mail: songzy@whu.edu.cn; hehe2013@whu.edu.cn

