## [Peer Review File · Nature Communications]

Reviewers' Comments:

Reviewer #1:

Remarks to the Author:

In this study, Dong et al. revealed that the EMC2-SLC25A46-Mic19 axis regulated ER-mitochondria contacts and that Mic9 was important for PA transport from the ER to mitochondria. Mic19 knockout induced UPRmt and ER stress in hepatocytes and triggered the NASH phenotype in mice. In general, this study provides novel insights into how PA is transported from the ER to mitochondria. However, there are some critical issues that should be addressed.

1. Figure 1 shows that Mic19 KO led to an increased distance between the ER and mitochondria. Did Mic19 KO affect the transport of other substrate from ER to mitochondria because of the longer distance between them, such as Ca²⁺.
2. Although the authors provide evidence that Mic19 did not bind with PC, PS and PE, these phospholipid species were also decreased in mitochondria of Mic19 KO mouse liver. This should be at least discussed.
3. Mic19 KO significantly decreased the protein levels of SLC25A46 and other subunits of the MICOS complex. What is the underlying mechanism? Are these effects of Mic19 KO dependent or independent of cardiolipin? Dose increased level of mitochondrial proteases play a role in this?
4. In addition to the level of respiratory chain complexes, mitochondrial oxidative phosphorylation of Mic19 KO hepatocytes needs to be analyzed by Seahorse.
5. Energy expenditure was greatly increased in Mic19 LKO mice; however, Mic19 KO impaired mitochondrial fatty acid β -oxidation in the mouse liver. The authors did not explain the contradiction. The Mic19 LKO mice showed lean phenotype and increased plasma FFA level. Was the lipolysis in adipose tissue been induced in Mic19 KO mice?
6. The data of liver weight and adipose tissue weight of Mic19 LKO mice should be presented.
7. Did Mic19 re-expression reverse the decreased body weight of Mic19 LKO mice?
8. In addition to Mic19 KO mice, the therapeutic effects of Mic19 overexpression in diet-induced NASH mice should also be explored.
9. Figure 6H shows increased Acc-1 and p-Acc1 levels, although the ratio of p-Acc1/Acc1 was unchanged. However, the authors described this result as "Acc1, p-Acc1 and Fasn, the major enzymes responsible for de novo fatty acid synthesis, were not changed in Mic19 mouse liver", which is incorrect.
10. The expression level of Cd36 in the livers of Mic19 LKO mice was ten times higher than that in control mice, which suggests that lipid uptake may contribute to Mic19 KO-induced lipid accumulation in the liver. The authors should at least discuss this.

Reviewer #2:

Remarks to the Author:

In the study by Dong et al. the authors propose that the Mic19-SLC25A46-EMC2 axis is mediating the transport of the phospholipid PA from the ER to the inner membrane of mitochondria and that the impairment of this pathway is linked to NASH and liver fibrosis. The model proposed is based on cell culture experiments, a liver-specific Mic19 knock-out mouse, and a MCD-induced NASH model. The authors suggest that Mic19 and SLC25A46 are involved in ER-mitochondrial contact site formation. Using protein-lipid overlay assays they show that Mic19 could bind PA and loss of Mic19 causes a reduction in the level of many phospholipids including PA, CL, PC, PS and PE. In the second part of the paper, they focused on determining the role of Mic19 in liver function by using the liver-specific Mic19 knockout mouse. They show that loss of Mic19 in the liver caused

induction of the mitochondrial unfolded protein response, impairment of mitochondrial fatty acids oxidation, accumulation of TAGs in the liver, and altered lipid metabolism and impairment of ketogenesis in these KO mice. Moreover, they found that hepatic levels of Mic19 and Mic60 are reduced in mouse models of MCD-induced non-alcoholic steatohepatitis. Overall, we feel that the manuscript provides a number of highly interesting observations and that in particular the description of the liver-specific Mic19-KO mouse with the observed phenotypes is of high quality and significance. Still, the mechanistic interpretations are not sufficiently supported by the data provided.

Major concerns:

1. We feel that some conclusions, in particular related to the proposed direct function of Mic19 and SLC25A46 in PA transport, are not supported sufficiently by the data provided. There is no biochemical evidence that Mic19 or SLC25A46 directly mediate the transport of PA, or between which membranes this occurs, and whether this is indeed specific to PA. Thus, although there is clear support for a role of Mic19 in mitochondrial lipid metabolism and physiological consequences in an in vivo model, the mechanistic explanation for this is still lacking.
2. The role of altered cristae morphology, described to be an essential function of the MICOS complex, is not sufficiently considered in the discussion of the model. Could alterations of cristae morphology per se impair synthesis of certain lipids? Are the effects on mitochondrial lipids including cardiolipin specific to Mic19? The fact that Mic19 loss also leads to the loss of other MICOS subunits such as Mic60 raises the question to which extent the effects are specific to Mic19 or whether this is linked to the MICOS complex as such. Also the roles of Mic26 and Mic27, to known apolipoproteins of the MICOS complex involved in cardiolipin metabolism, need to be considered. Overall, it is not clear whether Mic19 has a direct role in lipid transport although the hypothesis is attractive and certainly needs further investigation.

Specific other concerns:

3. The effects of Mic19 or SLC25A46 in determining the extent of ER-mitochondrial contact site formation (Fig. 1 and 2) are not convincing as the effects are quite minor. Moreover, in Fig 1, It is not stated how the distance between the ER and mitochondria in EM images was objectively calculated. What were the criteria used to define this distance as it can be different based on how empirically different person determine this distance? Therefore, it is also important that it is performed in a double-blind fashion. How were the mitochondrial traces in Fig 1E drawn and was this used for the quantification. We also feel that the alteration of mitochondrial morphology (e.g. more fragmentation of mitochondria leading to more but smaller mitochondria) could indirectly affect the quantification of apposition of mitochondria to the ER. The authors should normalize the e.g. the quantifications in Fig. 1G or Fig. 2K not to "number of mitochondria" but rather to "mitochondrial membrane surface/length".
4. In Fig 2A, the interaction between SLC25A46 and Mic19 is only shown by using highly overexpressed FLAG-SLC25A46. What about the interaction between the endogenous proteins? The molecular weights of the bands in the western blot are not indicated. There are double bands in the SLC25A46. Is it that the lower band corresponds to the endogenous SLC25A46. If yes, it is surprising that the endogenous SLC25A46 does not interact with FLAG version and is missing in decoration with SLC25A46. And instead in FLAG decoration a slightly higher molecular weight band appears. What does that mean? The co-IP also lacks decoration with a protein which does not bind as a negative control. At the same time, what about the other way around interaction using Mic19 antibody? It is important to show how Mic19 interacts with these components.
5. The lipidomics experiments for these phospholipids that are present both in ER and mitochondria are very tricky and difficult to analyse. Please show the purity of the mitochondrial fraction analysed for lipidomics, specifically with many ER and mitochondrial markers using western blots. In addition, how was the levels of these phospholipids in the individual fractions for MAM, mitochondria, and ER. Lower amounts of these phospholipids in mitochondria, does not mean that the transport is impaired. Also synthesis, or conversion to other lipids, or export back to other organelles, or intramitochondrial transport to inner membrane subcompartments can be altered. Similarly, the reduction was seen in many PLs, like PA, PS, CL etc. There is no explanation provided about how can these changes be explained based on the proposed role of Mic19 on PA transport alone. The levels of PLs are only represented in a relative manner but it would be also interesting to look at the absolute values of these PL and compare with the published literature.
6. NAO staining was used to further confirm the CL content in the cell culture model. However, it is

not the most convincing method to study CL levels as some papers suggest that the binding could be mitochondrial potential dependent. We think that lipidomics with the absolute levels of CL is more convincing way and should be performed here as well.

7. Fig. 3G and 3I: the labelling of the y-axis could be optimized.

8. The scheme in Fig 4 is very far-fetched. Just based on the overall PL data and binding data, one cannot conclude that Mic19 participates in PA transport from OMM to IMM. This schematic is gross misrepresentation of the data and should be adapted accordingly. (see major comments above). The data is not sufficient to show any role of Mic19 in transport of PA from OMM to IMM. A biochemical fractionation of pure OMM, IMM and ER membranes and corresponding lipidomics is required for this. If the transport from OMM to IMM is impaired, why would authors see the decrease in total levels of PA (and other PL). Accumulation of PA in the OMM is expected and needs to be tested.

9. What is the role of Mic19 binding to CL as shown in Fig 4BC?

10. The levels of cardiolipin synthesis enzyme are not altered. Yet, this does not mean that the CL synthesis is not affected and therefore the transport is impaired. This is again a very far-fetched interpretation and should be corrected. Instead, rates of CL synthesis should be directly analysed (e.g. in isolated mitochondria or using metabolomic flux analysis in vivo) and alternative explanations should be discussed at least.

11. Fig 5C does not convincingly show ER- mitochondrial contact sites. Are there more convincing micrographs? This should to be replaced and as stated in comment 3, check for the correctness of the method of quantification.

12. It is not explained, even in the discussions, how does Mic19 as a protein could bind PA and perform the transfer of PA. Is there a predicted lipid binding domain?

13. The characterisation of the Mic19 liver specific KO is interesting but how and why Mic19 liver specific KO show these phenotypes- please provide more about this in the discussion. Currently the discussion is very vague and does not include sufficient explanations about the results obtained.

Reviewer #3:

Remarks to the Author:

In the manuscript titled "Mic19 depletion impairs endoplasmic reticulum-mitochondrial phosphatidic acid transport and mitochondrial lipid metabolism and triggers liver disease," Dong and co-authors perform impressive phenotyping of Mic19 depleted cells and liver specific knockout mice. The authors find that Mic19 loss leads to decreased mitochondria-ER contact, disorganization of mitochondrial cristae junctions, decreased levels of mitochondrial ETC, and indicators of lowered FAO. These all correspond to a striking in vivo NASH phenotype. The authors propose that Mic19 acts to transport phosphatidic acid (PA) into mitochondria where it is needed for cardiolipin synthesis (CL), and that the phenotypes of Mic19 loss result from loss of this function. This is an impressive amount of work, and a beautiful phenotype. Unfortunately, the data are over interpreted and while suggestive, do not test the causality of the mechanism proposed by the authors.

Major points:

The only rescue experiment in the paper is re-expression of Mic19 in LKO mice. Thus, the only conclusion we can draw is that all the observed phenotypes, from molecular differences in phospholipid abundance, to overt fat accumulation in LKO livers, result from Mic19 loss. The phenotypes are correlated, but we cannot begin to put them into a pathway and say that one leads to another, yet the authors make these types of claims over and over again throughout the manuscript (e.g. in the abstract: "impairment of ER-mitochondria contacts by Mic19... ..causes reduction of CL" and "reduction of ER-mitochondrial contacts and disorganization of mitochondrial cristae result in mitochondrial unfolded protein stress response (UPRmt) in mouse hepatocytes, impairing liver mitochondrial fatty acid β -oxidation and lipid metabolism, which further spontaneously triggers nonalcoholic steatohepatitis (NASH)"). These unproven statements occur in every section of the paper. In the absence of further mechanistic experiments, the language in the manuscript must be edited to make clear that a causal relationship has not been established, other than that Mic19 loss results in each of these changes.

In particular, the authors purport that all the downstream effects they see of Mic19 knockout (e.g. membrane disorganization, ETC complexes, FAO, NASH etc) are a result of blocked PA transport into mitochondria and resulting decreased cardiolipin synthesis. However, the authors cannot conclude this from the data. Mic19 LKO mice show decreases in every phospholipid class in their mitochondria (Figure 3A-B), likely due to loss of MICOS as a whole (Figure 5A). Loss of MICOS likely contributes to membrane disorganization completely in the absence of changes in membrane lipid composition. While other mechanistic studies have linked CL to ETC function and mitoUPR in the past, there are no experiments herein that show the specificity of the CL effect on the downstream phenotypes in this study. To make these claims, the authors need to devise a way to specifically rescue CL vs. other mitochondrial lipids in this model. Again, in the absence of mechanistic experiments, which could easily be performed in their cell line systems, the language in the manuscript must be edited to make clear that a causal relationship between decreased mitochondrial CL and the other measured phenotypes has not been established.

The authors draw conclusions about fatty acid synthesis and oxidation from surrogate measures that do not directly assay these processes. In Figure 6, ACC1 expression is definitely increased in LKO mice. However, the authors only discuss that the pACC1/ACC1 ratio is not changed. I do not agree with the interpretation that this means fatty acid synthesis can be ruled out as unaffected. Similarly, changes in fatty acid oxidation are inferred from differences in gene expression that are only detectable in fasted mice. The authors need to provide functional assays for fatty acid synthesis and oxidation, or revise the language to be more suggestive and less conclusive.

How Mic19 is connected to changes in FAO gene expression is unclear. Do the authors propose it is through mitoUPR and/or ER stress signaling? If so, they need to test those claims using inhibition or genetic manipulation of those pathways to show their causal involvement, or make clear that this is a hypothesis.

In Figure 9, the authors show correlative changes in Mic19 and Mic60 expression in a model of NASH. There is nothing in this figure to suggest these changes are causal. The text must reflect this, or alternatively, the authors could perform rescue experiments with their Mic19 AAV in this model.

Minor points:

For all the mito-ER colocalization studies, what is the total change in mitochondrial surface area in the knockouts/knockdowns? It appears by eye that MICOS loss decreases mitochondrial abundance. If this is true, how is it taken into account? If there are fewer mitochondria, then won't there naturally be less ER-mitochondria contact?

In Figure 2, it appears that GAPDH is increased in the western blots, which exaggerates the degree to which SLC25A46 and Mic19 affect one another's protein levels. The language that is used to describe this in the text exaggerates this finding.

It would be helpful to have Figure S3A earlier, as the authors perform experiments in LKO mice prior to this figure.

Figure 5C is uninterpretable. The image is too poor quality to assess their conclusions, it is not possible to see cristae in the fl/fl control.

Figure S4I is unnecessary, this is a standard technique and it is unclear why the authors felt it was needed to have a schematic (as opposed to any other experiment in the paper).

The authors show convincing data indicating the accumulation of liver and serum FFA, and liver TG in LKO mice. They find decreases in some FAO transcripts, and conclude that this results in decreased fatty acid oxidation in LKO mice. They also show convincing decreases in ETC abundance in LKO mice. However, Figure S4 shows increased oxygen consumption and CO₂ production. Why? This is not well explained or connected to the rest of the data in the paper. Are the elevated serum FFA driving increased FAO and respiration in other tissues like muscle?

Reviewer #4:

Remarks to the Author:

In this paper, the authors examined the function of the MICOS subunit Mic19, at the molecular, cellular and physiological levels. The first part of the study focuses on the role of Mic19 and its partners, including SLC25A46, in ER-mitochondria contact regulation, transport and metabolism of mitochondrial lipids, such as PA, and protein-protein and protein-lipid interactions. For this, the authors used a variety of techniques ranging from super resolution microscopy and EM imaging, pull down assays from Hela control, Mic19 KO or shMic19, and shSLC25A46 cells. Lipidomics data from hepatic mitochondrial fractions of Mic19 liver-specific KO mice are also presented and biochemical assays with purified proteins are shown. The second part of the paper focuses on the pathophysiological consequences related to the absence of Mic19 in the liver. More specifically, the paper reports that Mic19 is required for mitochondrial membrane organization, and its absence leads to a number of mitochondrial defects (UPR(mt), oxidative phosphorylation impairment, decreased ATP production, increased ROS levels) and ER stress. Linked to this, Mic19 LKO mice show liver FA metabolism defects due to FA beta-oxidation. This leads to increased TG levels and hepatic steatosis, as well as liver inflammation and finally liver fibrosis. Of note, the authors have performed rescue experiments by re-expressing a FLAG-tagged version of Mic19, which reverse the phenotype described. Overall, the absence of Mic19 triggers liver disease (NASH, fibrosis) via mitochondria function impairment. In the last section of the manuscript, the authors trigger the liver disease (NASH) by a specific diet, and show that consequently Mic19 expression decreased in the liver, whereas it was shown before that Mic19 depletion was the cause of liver disease.

The second part of the paper is well executed and is impressive in terms of metabolic data collected, although I don't know what conclusion to draw from the MCD-induced fatty liver disease model, as Mic19 is certainly not the only protein whose expression level varies following such treatment. The paper is generally well written. The data are well put together and follow a logical order. My main concerns are with the first part of the paper and final conclusions about the mechanism by which Mic19 supposedly operates.

Major points

1. I'm not convinced by the decrease in colocalization between ER and mitochondria as presented in the figures 1A, 1B, 1C, 1D, 2F, 2G and quantification shows very modest effects at best. There is still a lot of MAMs. Therefore, the authors should moderate their claims: the regulatory effect on contact sites formation by SLC25A46 or Mic19 is minor. This suggests that the effects obtained latter in vivo are probably not due to a defect in the formation of contact sites between ER and mitochondria. By contrast, mitochondria are in a very bad state, and shape, in the absence of Mic19 compared to controls, this is obvious in fig1E. The differences in morphology (shorter, round) should be addressed with more emphasis. Are there less mitochondria in Mic19 LKO cells? Is there mitophagy occurring?

2. Lipidomics. First, it is unclear based on the data provided whether the authors have properly purified mitochondria from MAMs, ER and others organelles. No western blot showing the evidence is provided. Second, the problem is that not only PA or CL decreased in Mic19 LKO mitochondria, but also all analyzed lipids, including PC, PE and PS, which questions the specificity of the EMC2-SLC25A46-Mic19 axis with respect to PA (although this is the proposed model in fig 4F and S8).

Related to this, I disagree with the conclusion from figure 3 (ie. last sentence from the first paragraph page 5.) Since all lipids analyzed decreased, it is possibly not only a problem of CL synthesis. Moreover, no experiment showed that the trafficking of PA from the ER to mitochondria was affected. Mic19 is a subunit of MICOS at the OMM-IMM interface, not acting as a PA transporter between ER and mitochondria.

3. NAO. NAO is used at 10 μ M: it is known that at high concentration NAO fluorescence is quenched (PMID 12124423) especially in zones where NAO is confined such as in mitochondria. The green fluorescence of NAO may decrease in the presence of CL (PMID 24372165, 1396703)

and could also depend on membrane potential. There is no control from the authors that the drop of NAO fluorescence is due to CL decrease or due to something else (self-quenching, membrane potential). Given that NAO is used at high concentration (10 μ M), data should be interpreted with caution. The authors must perform these controls.

4. Biochemical assays (figure 4). There is no evidence that Mic19-His and SLC24A46-His are correctly purified based on the WB. Standards and SDS-PAGE images must be shown, and a more detailed protocol in Method section must be provided.

Protein-lipid overlay assays. Lipids in such arrays are not presented to the proteins as they should be, that is, in a lipid bilayer context. Lipid concentrations in spots are also much higher than in cell situation. No reliable conclusion can be made based on protein-lipid overlay assay. They may serve as pilot experiment at best.

Liposome sedimentation. There is little information of how the experiment was made : concentration of proteins and lipids used? Are liposomes made of pure lipids (PA or PC or PE or PS) (as indicated in fig 4D)? No image of SDS-PAGE is provided, but a WB, no quantification is shown. The experiment has been repeated?

SLC25A46 is a transmembrane protein and therefore cannot be used in lipid binding assays as with a soluble protein. I also have doubts about the authors' method of purification of SLC25A46. Does SLC25A46 get inserted into proteoliposomes?

The model shown in Figure 4F is not scientifically supported. The authors did not present any PA transfer data, but merely a crude PA binding assay that should be interpreted with caution, even concern for SLC25A46. Therefore, the claim "Together, the EMC2-SLC25A46-Mic19 axis regulates ER-mitochondria contacts and participates in ER-mitochondrial PA transport" (First sentence in page 6) is not supported by the data: the effects on ER-mitochondria contacts are minor and PA transport is not analyzed.

5. Link between ER-mitochondria contact/PA transport and metabolic defects. Page 7: "Taken together, Mic19 LKO-caused reduction of ER-mitochondria contacts leads to UPRmt and ER stress". Discussion: the "impairment of PA transport is involved in the development and progression of liver diseases". These bold claims describe a correlation that has been made experimentally by the authors, but are not demonstrated as one causing the other. MICOS is one of the major structural elements of mitochondria that participate in many processes, such as generation of mitochondrial architecture, protein import into mitochondria, and lipid metabolism. Therefore, destabilization of MICOS by the absence of Mic19 may be the source of other problems than lipid transfer, suggesting that the metabolic issues observed here are possibly not related to PA transfer.

6. Rationale of figure 9. The authors treated mice to induce liver disease. They observed that the expression level of Mic19 decreases (as perhaps dozens of other proteins?) and concluded that their results are consistent with the function of Mic19 in repressing mouse liver disease. There is a problem of cause and consequence here that the authors do not explain.

Other points

a. Page 2: "how PA transport between ER and mitochondria is still unknown". I assume the authors meant "how PA is transported...". In fact there are several recent papers that describe mechanisms driven by lipid-transfer proteins to transport PA between the ER and mitochondria; they should be cited here (For example PMID 33938112, 36693319, 30093493, 36282247).

b. Is there any evidence that EMC2 and SLC25A46 promote MAM formation through direct physical interactions? In the introduction, the authors state this as a fact, although I'm not sure whether this is actually described. In the paper (ref#36) cited by the authors, EMC2 was found among many others proteins in a bioID screen, which is not interaction per se, but proximity. See also the first sentence of second paragraph in page 4.

c. In figure 3B, data are presented as fold change versus control mice. Then why values for lipids

from control mice (Mic19flox/flox) are in red color in the heatmap (which suggest more lipids compared to control)?

d. in vivo experiments from fig5A, B are convincing, but TEM images in fig 5C are not clear and of poor quality. What are we supposed to see? Authors should identify organelles and contact between them. Cristea number are quantified in fig 5E but no cristae are seen in fl/fl condition from fig5C.

e. The sentence "the levels of liver and serum free fatty acids were dramatically elevated in Mic19 LKO mice (Figures 6F and 6G)." is written in the text in page 7 and again in page 8.

We sincerely thank the reviewers for their valuable comments and constructive criticisms, which were of great help in revising the manuscript. According to the reviewers' comments and suggestions, the revised manuscript has been systematically improved with the new data and additional interpretations. Reviewers' comments are responded point-by-point as below. Reviewers' points are underlined and italic for easier reference. Descriptions for all newly performed experimental results and other changes are highlighted in blue in the revised manuscript.

Reviewer 1

In this study, Dong et al. revealed that the EMC2-SLC25A46-Mic19 axis regulated ER-mitochondria contacts and that Mic9 was important for PA transport from the ER to mitochondria. Mic19 knockout induced UPRmt and ER stress in hepatocytes and triggered the NASH phenotype in mice. In general, this study provides novel insights into how PA is transported from the ER to mitochondria. However, there are some critical issues that should be addressed.

1. Figure 1 shows that Mic19 KO led to an increased distance between the ER and mitochondria. Did Mic19 KO affect the transport of other substrate from ER to mitochondria because of the longer distance between them, such as Ca²⁺.

We greatly appreciate the reviewer's comments and suggestions. According to the reviewer's comments, we used live-cell confocal microscopy to image mitochondrial calcium using the mitochondria-targeted genetically encoded calcium sensor Mito-R-GECO1 (Wu et al., 2013). Briefly, control or Mic19 KO HeLa cells expressing Mito-R-GECO1 and Mito-Green (a mitochondrial marker, green) were imaged by confocal microscopy (Figure R1A, shown in below). And we measured the Ca²⁺ levels in mitochondria by flow cytometric analysis as well (Figure R1B, shown in below). These data indicate that loss of Mic19 decreases the Ca²⁺ levels in mitochondria. Therefore, Mic19 depletion-caused decrease of ER-mitochondria contacts also impairs Ca²⁺ transfer between ER to mitochondria. Because the data about Ca²⁺ levels are not the main line of our manuscript, and the length of article in "Nature Communications" is also limited. Therefore, we will provide Figure R1 and the other related data in our future paper.

Reference

Wu, J., Liu, L., Matsuda, T., Zhao, Y., Rebane, A., Drobizhev, M., Chang, Y.F., Araki, S., Arai, Y., March, K., et al. (2013). Improved orange and red Ca^{2±} indicators and photophysical considerations for optogenetic applications. ACS chemical neuroscience 4, 963-972.

Figure R1. Mitochondrial level of Ca²⁺ in cells. (A) Mitochondrial calcium levels in live WT or Mic19 KO HeLa cells expressing mitochondrial-matrix targeted calcium sensor Mito-R-GECO1 and Mito-Green with representative confocal images. At least three independent experiments were performed. (B) Mitochondrial calcium levels in live control or Mic19 KO HeLa cells expressing mitochondrial-matrix targeted calcium sensor Mito-R-GECO1 and Mito-Green were measured by flow cytometric analysis. 3 independent experiments were performed. Data are presented as mean \pm SD, statistical significance was assessed by Student's t-test, *** $p < 0.001$.

2. Although the authors provide evidence that Mic19 did not bind with PC, PS and PE, these phospholipid species were also decreased in mitochondria of Mic19 KO mouse liver. This should be at least discussed.

We appreciate the reviewer's comments and suggestions. Our results indicate that Mic19 did not interact with PC, PS and PE, but PC, PS and PE were also reduced in mitochondria of Mic19 KO cells (Figure 3B of the revised manuscript). It is probably that Mic19 depletion led to a reduction in the ER-mitochondria contacts and accompanied by an increased distance between these two organelles (Figure 1 of the revised manuscript). Since PC and PS can also transport from ER to mitochondria, and the Mic19 KO-caused the reduction of ER-mitochondria contacts may impair the transport of PC and PS to mitochondria. In addition, PS is the precursor for PE, the decreased PE may be the consequence of the reduced PS. Similarly, previous studies have reported that Mfn2 knockout also causes a reduction of PS, PC and PE in liver mitochondria although Mfn2 just binds to PS (Hernández-Alvarez et al., 2019). We have discussed this issue in the revised manuscript.

Reference

Hernández-Alvarez, M.I., Sebastián, D., Vives, S., Ivanova, S., Bartoccioni, P., Kakimoto, P., Plana, N., Veiga, S.R., Hernández, V., Vasconcelos, N., et al. (2019). Deficient Endoplasmic Reticulum-Mitochondrial Phosphatidylserine Transfer Causes Liver Disease. *Cell* 177, 881-895.e817.

3. *Mic19 KO significantly decreased the protein levels of SLC25A46 and other subunits of the MICOS complex. What is the underlying mechanism? Are these effects of Mic19 KO dependent or independent of cardiolipin? Dose increased level of mitochondrial proteases play a role in this?*

According to the reviewer's comments, we investigated the underlying mechanism of SLC25A46 degradation. We treated cells with cycloheximide (CHX, protein synthesis inhibitor). CHX treatment caused significant reduction (degradation) of SLC25A46 in control and Mic19 KO cells (Figures S2A and S2B of the revised manuscript). However, further MG132 treatment could not inhibit CHX-induced reduction of SLC25A46 in control and Mic19 KO cells (Figures S2C and S2D of the revised manuscript), indicating that the degradation of SLC25A46 is independent on the ubiquitin-proteasome pathway. Then, we explored the effect of mitochondrial proteases on Mic19 KO-induced the degradation of SLC25A46. We infected cells with control or shMic19 lentivirus in WT, OMA1 KO, or Yme1L KO cells. 5 days later, cell lysates were analyzed by Western blotting against SLC25A46. Western blotting analysis revealed that the depletion of OMA1 or Yme1L led to a significant inhibition of SLC25A46 degradation in Mic19 KO cells (Figures S2E and S2F of the revised manuscript), indicating that mitochondrial protease OMA1 and Yme1L contribute to Mic19 KO-caused the degradation of SLC25A46. Additionally, we performed knocked down of CLS1 that is important for cardiolipin synthesis in cells. Western Blotting analysis showed that CLS1 knockdown decreased the protein levels of Mic19, Mic60 and SLC25A46 in cells (Figures S2G and S2H of the revised manuscript). These results suggest that the degradation of SLC25A46 and MICOS subunits is probably cardiolipin-dependent. We have provided these data in the revised manuscript.

4. *In addition to the level of respiratory chain complexes, mitochondrial oxidative phosphorylation of Mic19 KO hepatocytes needs to be analyzed by Seahorse.*

As suggested by the reviewer, we analyzed oxygen consumption rates of 3-month-old Mic19^{flox/flox} and Mic19 LKO mouse liver cells by high-resolution respirometry with Oroboros O2k system (the function is similar to the Seahorse), and the data were shown in Figures 5H and S5C-S5H of the revised manuscript. At the onset of the symptomatic stage, ADP stimulated

respiration [i.e., oxidative phosphorylation (OXPHOS)], which reflected the maximal capacity to generate ATP and uncoupled respiration, providing the maximal electron transfer (ET) capacity. Respiration from cytochrome c reflected Integrity of mitochondrial membrane. As expected, that the oxygen consumption was significantly decreased when using CI substrates (i.e., CI OXPHOS) or CI and CII substrates simultaneously in phosphorylating and uncoupled states (i.e., CI + CII OXPHOS and ET) in Mic19 LKO mice. Similarly, the respiration from cytochrome c oxidase (i.e., CIV) was obviously reduced in Mic19 LKO mice as well. In conclusion, these data suggest that Mic19 LKO caused the defect of mitochondrial oxidative phosphorylation in mouse liver cells.

5. Energy expenditure was greatly increased in Mic19 LKO mice; however, Mic19 KO impaired mitochondrial fatty acid β -oxidation in the mouse liver. The authors did not explain the contradiction. The Mic19 LKO mice showed lean phenotype and increased plasma FFA level. Was the lipolysis in adipose tissue been induced in Mic19 KO mice?

We appreciate the reviewer's comments and suggestions. Mic19 LKO mice displayed remarkably increased oxygen consumption (VO₂) and carbon dioxide production (VCO₂) rates by mice metabolism cage experiment (Figures S6K-S6N of the revised manuscript). As adipose tissue is an important organ in regulating energy balance, serving not only as an energy store but also as a modulator of metabolism. We found that the weight of adipose tissues including eWAT (epididymal white adipose tissue) and iWAT (inguinal white adipose tissue) were slightly decreased (Figure S6O of the revised manuscript). However, qRT-PCR analysis revealed that the mRNA levels of some mitochondrial β -oxidation genes including Cpt1b, Cpt2 and Acads were significantly increased in Mic19 LKO mouse iWAT (Figure S6P of the revised manuscript). These results indicate that lipolysis is increased in the iWAT of Mic19 LKO mice, which may be a compensatory response of the impairment of β -oxidation in the liver tissue, thus increasing the energy expenditure in Mic19 LKO mice. In addition, Mic19 LKO-increased lipolysis in the iWAT may cause lean phenotype and the increased plasma FFA level in Mic19 LKO mice.

6. The data of liver weight and adipose tissue weight of Mic19 LKO mice should be presented.

According to the reviewer's suggestion, we detected the weight of liver, eWAT (epididymal white adipose tissue) and iWAT (inguinal white adipose tissue) from 3-month-old Mic19^{fl_{ox}/fl_{ox}} or Mic19 LKO mice. The ratio of liver to body weight, which is extensively used as a clinical parameter to assess liver enlargement, was significantly increased (Figure S6O of the revised manuscript). While the ratio of WAT to body weight was slightly decreased in Mic19 LKO mice

compared to control mice (Figures S6O of the revised manuscript). These data have been provided in the revised manuscript.

7. Did Mic19 re-expression reverse the decreased body weight of Mic19 LKO mice?

According to the reviewer's comments, we have measured the body weight in 3-month-old or 7-month-old Mic19^{flox/flox}, Mic19 LKO or Mic19 re-expressed Mic19 LKO mice. Mic19 re-expression remarkably reversed the decreased body weight of Mic19 LKO mice (Figures S9A and S9B of the revised manuscript).

8. In addition to Mic19 KO mice, the therapeutic effects of Mic19 overexpression in diet-induced NASH mice should also be explored.

As suggested by the reviewer, we investigated the effect of Mic19 overexpression on MCD-induced liver disease. adeno-associated virus encoding Mic19-Flag were injected by a tail vein into 8-week-old MCD mice. After overexpression of Mic19-Flag in MCD-treated mouse liver, the upregulation of TG level and mRNA level of inflammation-related genes *Cd68*, *Cxcl10* and *Tnf* were suppressed and recovered to be similar to that of normal diet mice (Figures S9G-S9I). Moreover, H&E staining revealed that Mic19-Flag overexpression remarkably suppressed MCD-caused hepatic steatosis in mice (Figure S9J). Therefore, Mic19 overexpression suppresses MCD-induced NASH.

9. Figure 6H shows increased Acc1 and p-Acc1 levels, although the ratio of p-Acc1/Acc1 was unchanged. However, the authors described this result as "Acc1, p-Acc1 and Fasn, the major enzymes responsible for de novo fatty acid synthesis, were not changed in Mic19 mouse liver", which is incorrect.

We sincerely thank the reviewer's comments. In the de novo synthesis pathway of lipids, acetyl-CoA is firstly carboxylated to form malonyl-CoA. This is followed by a series of oxidation reactions that ultimately result in the production of triglycerides. The key enzymes involved in this process are Acc1, p-Acc1, and Fasn. Acc1 participates in the synthesis of malonyl-CoA, while p-Acc1 inhibits the activity of Acc1 (Ha et al., 1994). Fasn primarily catalyzes the formation of palmitic acid esters from acetyl-CoA and malonyl-CoA in the presence of NADPH, which are then used for the synthesis of long-chain saturated fatty acids. Therefore, we detected these protein levels by Western blotting. Although both Acc1 and p-Acc1 in the liver of Mic19 LKO mice were upregulated, the ratio between the two enzymes showed no significant difference. Furthermore, there was also no significant difference in Fasn between control and

Mic19 LKO mouse liver. Therefore, we described that “p-Acc1/Acc1 (the ratio) and Fasn, the major enzymes responsible for de novo fatty acid synthesis, were not changed in Mic19 mouse liver” in the revised manuscript.

Reference

Ha, J., Daniel, S., Broyles, S.S., and Kim, K.H. (1994). Critical phosphorylation sites for acetyl-CoA carboxylase activity. *The Journal of biological chemistry* 269, 22162-22168.

10. The expression level of Cd36 in the livers of Mic19 LKO mice was ten times higher than that in control mice, which suggests that lipid uptake may contribute to Mic19 KO-induced lipid accumulation in the liver. The authors should at least discuss this.

We greatly appreciate the reviewer’s suggestion. We have provided the sentence “the mRNA level of CD36 (fatty acid transporter, a marker of fatty liver disease) of Mic19 LKO mouse liver (3 months old) was more than 10 times that of control mouse liver under fed or fasted conditions (Figure S7A), indicating that lipid uptake may contribute to Mic19 LKO-induced lipid accumulation in mouse liver.” in the revised manuscript.

Reviewer #2

In the study by Dong et al. the authors propose that the Mic19-SLC25A46-EMC2 axis is mediating the transport of the phospholipid PA from the ER to the inner membrane of mitochondria and that the impairment of this pathway is linked to NASH and liver fibrosis. The model proposed is based on cell culture experiments, a liver-specific Mic19 knock-out mouse, and a MCD-induced NASH model. The authors suggest that Mic19 and SLC25A46 are involved in ER-mitochondrial contact site formation. Using protein-lipid overlay assays they show that Mic19 could bind PA and loss of Mic19 causes a reduction in the level of many phospholipids including PA, CL, PC, PS and PE. In the second part of the paper, they focused on determining the role of Mic19 in liver function by using the liver-specific Mic19 knockout mouse. They show that loss of Mic19 in the liver caused induction of the mitochondrial unfolded protein response, impairment of mitochondrial fatty acids oxidation, accumulation of TAGs in the liver, and altered lipid metabolism and impairment of ketogenesis in these KO mice. Moreover, they found that hepatic levels of Mic19 and Mic60 are reduced in mouse models of MCD-induced non-alcoholic steatohepatitis. Overall, we feel that the manuscript provides a number of highly interesting observations and that in particular the description of the liver-specific Mic19-KO mouse with the observed phenotypes is of high quality and significance. Still, the mechanistic interpretations are not sufficiently supported by the data provided.

Major concerns:

1. We feel that some conclusions, in particular related to the proposed direct function of Mic19 and SLC25A46 in PA transport, are not supported sufficiently by the data provided. There is no biochemical evidence that Mic19 or SLC25A46 directly mediate the transport of PA, or between which membranes this occurs, and whether this is indeed specific to PA. Thus, although there is clear support for a role of Mic19 in mitochondrial lipid metabolism and physiological consequences in an in vivo model, the mechanistic explanation for this is still lacking.

We sincerely thank the reviewer for his/her comments. According to the reviewer's comments, we performed liposome extraction and phospholipid transfer assay in vitro, separately. The data reveal that Mic19 or SC25A46 are able to bind to and transfer PA in vitro (Figures 4F, 4G, 4J, 4K, S4E and S4F of the revised manuscript).

2. The role of altered cristae morphology, described to be an essential function of the MICOS complex, is not sufficiently considered in the discussion of the model. Could alterations of cristae morphology per se impair synthesis of certain lipids? Are the effects on mitochondrial lipids including cardiolipin specific to Mic19? The fact that Mic19 loss also leads to the loss of other MICOS subunits such as Mic60 raises the question to which extent the effects are specific to Mic19 or whether this is linked to the MICOS complex as such. Also the roles of Mic26 and Mic27, to known apolipoproteins of the MICOS complex involved in cardiolipin metabolism, need to be considered. Overall, it is not clear whether Mic19 has a direct role in lipid transport although the hypothesis is attractive and certainly needs further investigation.

We greatly appreciate and agree to the reviewer's comments and suggestions. According to the reviewer's comments, we performed liposome extraction and phospholipid transfer assay in vitro, separately. The data reveal that Mic19 or SC25A46 are able to bind to and transfer PA in vitro (Figures 4F, 4G, 4J, 4K, S4E and S4F of the revised manuscript).

Additionally, Mic19 deletion results in a reduction of the MICOS complex subunits, including Mic60 (Figures 5A and 5B of the revised manuscript), Mic26 and Mic27 (Guarani et al., 2015). Mic60 could co-sediment with liposomes and regulates membrane shaping (Zhou et al., 2022), and Mic26 or Mic27 (the subunit of MICOS complex) are also associated with cardiolipin metabolism (Anand et al., 2020). Moreover, the alterations of inner membrane morphology may be linked to membrane lipid metabolism and synthesis (Tasseva et al., 2013), thus, the alterations of inner membrane morphology may potentially influence the localization of enzymes and proteins on the mitochondrial inner membrane and impair phospholipid synthesis. Therefore, Mic19 may directly or indirectly regulate lipid metabolism and transport. In this manuscript, we focus on the role of Mic19 on PA binding and transport. We have discussed the issues in the revised manuscript.

References

- Anand, R., Kondadi, A.K., Meisterknecht, J., Golombek, M., Nortmann, O., Riedel, J., Peifer-Weiß, L., Brocke-Ahmadinejad, N., Schlütermann, D., Stork, B., et al. (2020). MIC26 and MIC27 cooperate to regulate cardiolipin levels and the landscape of OXPHOS complexes. *Life science alliance* 3, e202000711.
- Guarani, V., McNeill, E.M., Paulo, J.A., Huttlin, E.L., Fröhlich, F., Gygi, S.P., Van Vactor, D., and Harper, J.W. (2015). QIL1 is a novel mitochondrial protein required for MICOS complex stability and cristae morphology. *eLife* 4, e06265
- Tasseva, G., Bai, H.D., Davidescu, M., Haromy, A., Michelakis, E., and Vance, J.E. (2013). Phosphatidylethanolamine deficiency in Mammalian mitochondria impairs oxidative phosphorylation and alters mitochondrial morphology. *The Journal of biological chemistry* 288, 4158-4173.
- Zhou, J., Duan, M., Wang, X., Zhang, F., Zhou, H., Ma, T., Yin, Q., Zhang, J., Tian, F., Wang, G., et al. (2022). A feedback loop engaging propionate catabolism intermediates controls mitochondrial morphology. *Nature cell biology* 24, 526-537.

Specific other concerns:

3. The effects of Mic19 or SLC25A46 in determining the extent of ER-mitochondrial contact site formation (Fig. 1 and 2) are not convincing as the effects are quite minor. Moreover, in Fig 1, It is not stated how the distance between the ER and mitochondria in EM images was objectively calculated. What were the criteria used to define this distance as it can be different based on how empirically different person determine this distance? Therefore, it is also important that it is performed in a double-blind fashion. How were the mitochondrial traces in Fig 1E drawn and was this used for the quantification. We also feel that the alteration of mitochondrial morphology (e.g. more fragmentation of mitochondria leading to more but smaller mitochondria) could indirectly affect the quantification of apposition of mitochondria to the ER. The authors should normalize the e.g. the quantifications in Fig. 1G or Fig. 2K not to “number of mitochondria” but rather to “mitochondrial membrane surface/length”.

We greatly appreciate the reviewer’s comments and suggestions. We used ImageJ software to determine specific lengths using a scale and calculated the closest distance between mitochondria and the ER, defined as the minimum distance. When this distance was less than 40 nm, we considered it as evidence of interaction between mitochondria and the ER. Additionally, both the acquisition of electron microscope images was randomly performed and the statistical analysis was performed in a double-blind fashion. In Figure 1E, the mitochondrial traces were manually drawn using the pencil tool in Adobe Illustrator for easier visualization. Additionally, we measured the mitochondrial length of TEM images by ImageJ software, and the length of mitochondria in control or Mic19 KO cells was provided in Figures 1F and 2J of the revised manuscript. Moreover, according to the reviewer’s suggestion, we normalized “the number of ER-mito contacts” to “10 μ m mitochondrial membrane surface”, and we provided the new statistical data in Figures 1I, 2M of the revised manuscript.

4. In Fig 2A, the interaction between SLC25A46 and Mic19 is only shown by using highly overexpressed FLAG-SLC25A46. What about the interaction between the endogenous proteins? The molecular weights of the bands in the western blot are not indicated. There are double bands in the SLC25A46. Is it that the lower band corresponds to the endogenous SLC25A46. If yes, it is surprising that the endogenous SLC25A46 does not interact with FLAG version and is missing in decoration with SLC25A46. And instead in FLAG decoration a slightly higher molecular weight band appears. What does that mean? The co-IP also lacks decoration with a protein which does not bind as a negative control. At the same time, what about the other way

around interaction using Mic19 antibody? It is important to show how Mic19 interacts with these components.

In original Figure 2A, the lower band corresponds to the endogenous SLC25A46, which does not bind to Flag-SLC25A46 probably because endogenous SLC25A46 have high affinity to interact with endogenous SLC25A46. After long exposure of Western blotting, a slight level of endogenous SLC25A46 was precipitated by Flag-SLC25A46 (Figure R2, shown in below). To avoid the misunderstanding, we deleted the Western blot obtained by using anti-SLC25A46 in the Figure 2A of the revised manuscript. In addition, according to the reviewer's comments, we further validate the interaction between SLC25A46 and EMC2 by co-IP using anti-GFP (negative control) or anti-SLC25A46 (no good commercial anti-Mic19 for co-IP). Co-IP showed that endogenous SLC25A46 could interact with both Mic19 and EMC2 (Figure 2B of the revised manuscript). The new data were provided in Figure 2B of the revised manuscript.

Figure R2. The Western blot using anti-SLC25A46 (long exposure). 293T cells were transiently transfected with empty vector (control) or vector coding for Flag-SLC25A46. After 36h transfection, cell lysates were used for co-immunoprecipitation with anti-Flag M2 affinity gel at 4°C overnight, followed by SDS-PAGE and Western blotting analysis with anti-SLC25A46 antibodies.

5. The lipidomics experiments for these phospholipids that are present both in ER and mitochondria are very tricky and difficult to analyse. Please show the purity of the mitochondrial fraction analysed for lipidomics, specifically with many ER and mitochondrial markers using western blots. In addition, how was the levels of these phospholipids in the individual fractions for MAM, mitochondria, and ER. Lower amounts of these phospholipids in mitochondria, does not mean that the transport is impaired. Also synthesis, or conversion to other lipids, or export back to other organelles, or intramitochondrial transport to inner membrane subcompartments can be altered. Similarly, the reduction was seen in many PLs, like PA, PS, CL etc. There is no explanation provided about how can these changes be explained based on the proposed role of Mic19 on PA transport alone. The levels of PLs are only represented in a relative manner but it would be also interesting to look at the absolute values of these PL and compare with the published literature.

We greatly appreciate the reviewer's comments and suggestions. According to the reviewer's suggestion, we analyzed the purity of the mitochondrial fraction used for lipidomics analysis by Western blotting. Western blotting analysis showed mitochondrial marker Tom40 existed only in the fraction of mitochondria but not ER or MAM (mitochondria-associated membrane) (Figure S3C of the revised manuscript), indicating that mitochondrial fraction analyzed for lipidomics was pure.

In addition, it should be noted that it is very hard to get the enough amount of MAM for lipidomics analysis, and we agree the reviewer's comment that phospholipids could transport back to ER or to other organelles. Therefore, it is hard to conclude the direct role of Mic19 in PA transport. To further investigate the role of Mic19 in PA transport, we performed liposome extraction and phospholipid transfer assay in vitro, separately. The data reveal that Mic19 or SC25A46 are able to bind to and transfer PA in vitro (Figures 4F, 4G, 4J, 4K, S4E and S4F of the revised manuscript).

Because we performed a non-targeted lipidomics analysis, thus the content of PLs in the article is displayed with the relative values. And compared to target lipid analysis, non-target lipidomics can provide more types of lipids and specific lipid species names which include more comprehensive information. At the same time, it can also satisfy the need for comparing differences between groups.

Our results indicate that Mic19 did not interact with PC, PS and PE, but PC, PS and PE were also reduced in mitochondria of Mic19 KO cells (Figure 3B of the revised manuscript). It is probably that Mic19 depletion led to a reduction in the ER-mitochondria contacts and accompanied by an increased distance between these two organelles (Figure 1 of the revised manuscript). Since PC and PS can also transport from ER to mitochondria, and the Mic19 KO-caused the reduction of ER-mitochondria contacts may impair the transport of PC and PS to mitochondria. In addition, PS is the precursor for PE, the decreased PE may be the consequence of the reduced PS. Similarly, previous studies have reported that Mfn2 knockout also causes a reduction of PS, PC and PE in liver mitochondria although Mfn2 just binds to PS (Hernández-Alvarez et al., 2019). We have discussed this issue in the revised manuscript.

Reference

Hernández-Alvarez, M.I., Sebastián, D., Vives, S., Ivanova, S., Bartoccioni, P., Kakimoto, P., Plana, N., Veiga, S.R., Hernández, V., Vasconcelos, N., et al. (2019). Deficient Endoplasmic Reticulum-Mitochondrial Phosphatidylserine Transfer Causes Liver Disease. *Cell* 177, 881-895.e817.

6. NAO staining was used to further confirm the CL content in the cell culture model. However, it is not the most convincing method to study CL levels as some papers suggest that the binding could be mitochondrial potential dependent. We think that lipidomics with the absolute levels of CL is more convincing way and should be performed here as well.

We sincerely thank the reviewer's suggestions. We further performed lipidomics analysis for cardiolipin level in control, Mic19 KO or SLC25A46 knockdown (KD) cells. Compared to the control cells, Mic19 KO and SLC25A46 KD cells displayed a reduced cardiolipin level (Figures 3C-3F of the revised manuscript).

Additionally, we analyzed mitochondrial membrane potential by performing TMRM staining on control, Mic19 KO and SLC25A46 KD cells and then measuring the fluorescence intensity using flow cytometry. Both Mic19 KO and SLC25A46 KD decreased mitochondrial membrane potential in cells (Figures S3H and S3I of the revised manuscript).

7. Fig. 3G and 3I: the labelling of the y-axis could be optimized.

We thank the reviewer's comments. We have adjusted the labelling of y-axis of Figures 3G and 3I in the revised manuscript.

8. The scheme in Fig 4 is very far-fetched. Just based on the overall PL data and binding data, one cannot conclude that Mic19 participates in PA transport from OMM to IMM. This schematic is gross misrepresentation of the data and should be adapted accordingly. (see major comments above). The data is not sufficient to show any role of Mic19 in transport of PA from OMM to IMM. A biochemical fractionation of pure OMM, IMM and ER membranes and corresponding lipidomics is required for this. If the transport from OMM to IMM is impaired, why would authors see the decrease in total levels of PA (and other PL). Accumulation of PA in the OMM is expected and needs to be tested.

We greatly appreciate the reviewer's comments and suggestions. Because it is very hard to get the enough amount and pure fractions of OMM and IMM for lipidomics analysis; moreover, the processes for purification of the OMM and IMM fractions may dramatically impair phospholipids of OMM and IMM. Therefore, to further investigate the role of Mic19 in PA transport, we performed phospholipid transfer assay in vitro. phospholipid transfer assay indicates that Mic19 or SLC25A46 was able to participate in PA transfer directly in vitro (Figures 4G, 4K and S4F of the revised manuscript). In addition, we previously reported that Mic19 locates in mitochondrial inner membrane space (Tang et al., 2020), therefore, Mic19 may participate in PA transport from OMM to IMM. We have discussed the issue in the revised

manuscript.

Reference

Tang, J., Zhang, K., Dong, J., Yan, C., Hu, C., Ji, H., Chen, L., Chen, S., Zhao, H., and Song, Z. (2020). Sam50-Mic19-Mic60 axis determines mitochondrial cristae architecture by mediating mitochondrial outer and inner membrane contact. *Cell death and differentiation* 27, 146-160.

9. What is the role of Mic19 binding to CL as shown in Fig 4BC?

Mic19 is a key component of the MICOS complex, which plays an important role in maintaining the structure and function of the mitochondrial inner membrane. In our opinion, the interaction between Mic19 and cardiolipin may contribute to the preservation of the cristae structure of mitochondria. Moreover, cardiolipin is involved in many essential biochemical processes in mitochondria, including the respiratory chain and oxidative phosphorylation. Therefore, the interaction between Mic19 and cardiolipin may conduce to ensure the mitochondrial related physiological processes. In conclusion, the interaction between Mic19 and cardiolipin may play a crucial role in maintenance of mitochondrial cristae and the related functions. We have discussed the issue in the revised manuscript.

10. The levels of cardiolipin synthesis enzyme are not altered. Yet, this does not mean that the CL synthesis is not affected and therefore the transport is impaired. This is again a very far-fetched interpretation and should be corrected. Instead, rates of CL synthesis should be directly analysed (e.g. in isolated mitochondria or using metabolimic flux analysis in vivo) and alternative explanations should be discussed at least.

We greatly appreciate and agree to the reviewer's comments. Although the protein level of CLS1 or TAZ was not decreased, the cardiolipin level was significantly reduced in Mic19 or SLC25A46 depleted cells (Figures 3C-3J of the revised manuscript). Even if the protein level remains unchanged, the activity of phospholipid synthase may be inhibited, including enzyme inactivation modifications. Additionally, substrate limitation may be a possible factor. If the level of substrates needed for phospholipids synthesis decreases, the phospholipid production would be reduced. Furthermore, the impaired ER-mitochondria PA transport may lead to the reduced CL production in mitochondria. In conclusion, although the protein levels of CLS1 and TAZ were unchanged, there are many other factors that might lead to a reduction in cardiolipin level. We found that Mic19 or SLC25A46 is involved in the regulation of ER-mitochondria contacts (Figures 1 and 2 of the revised manuscript). As the precursor lipid of cardiolipin, mitochondrial PA is mainly transported from the ER. Therefore, we speculate that Mic19 or SLC25A46 depletion may impair ER-mitochondrial PA transport, leading to the reduced CL

production. We have discussed the issue in the revised manuscript.

11. Fig 5C does not convincingly show ER- mitochondrial contact sites. Are there more convincing micrographs? This should to be replaced and as stated in comment 3, check for the correctness of the method of quantification.

According to the reviewer's comments, we performed the electron microscope analysis of mouse liver tissue and provided the new images in Figure 5C of the revised manuscript. The red arrows showed the ER-mitochondrial contact sites. In addition, we measured the mitochondrial membrane surface of EM images by ImageJ software, and then normalized "the number of ER-mito contacts" to "10 μm mitochondrial membrane surface". The new statistical data were provided in Figure 5E of the revised manuscript.

12. It is not explained, even in the discussions, how does Mic19 as a protein could bind PA and perform the transfer of PA. Is there a predicted lipid binding domain?

Through various liposome-related experiments, we confirmed that Mic19 can bind to PA and participate in PA transport (Figure 4 of the revised manuscript). In addition, DisoLipPred and InterPro (2 online tools) predictions show that, beside to contain coil-coil and CHCH domain, Mic19 protein also contains some disordered lipid binding regions (Figure S4A of the revised manuscript). We have provided these data and the related description in the revised manuscript.

13. The characterisation of the Mic19 liver specific KO is interesting but how and why Mic19 liver specific KO show these phenotypes- please provide more about this in the discussion. Currently the discussion is very vague and does not include sufficient explanations about the results obtained.

As suggested by the reviewer, we provided more discussion about the possible mechanism for the characterisation of the Mic19 liver specific KO mice in the revised manuscript. The words are: "Mic19 depletion leads to the decreased ER-mitochondria contacts and impairs PA transport from ER to mitochondria, impairing CL metabolism and mitochondrial membrane disorganization (including the reduction of mitochondrial cristae and loss of mitochondrial cristae junctions) and UPR^{mt}, then causing mitochondrial dysfunctions such as weakened mitochondrial respiration, reduced ATP production and increased ROS levels, etc. (Figures 1, 3, 4, 5 and S5). As mitochondrion is a crucial organelle for fatty acid oxidation, its dysfunction also impairs mitochondrial β -oxidation, leading to hepatic lipid accumulation, NASH and liver fibrosis in Mic19 LKO mice (Figures 6 and 7)".

Reviewer #3

In the manuscript titled “Mic19 depletion impairs endoplasmic reticulum-mitochondrial phosphatidic acid transport and mitochondrial lipid metabolism and triggers liver disease,” Dong and co-authors perform impressive phenotyping of Mic19 depleted cells and liver specific knockout mice. The authors find that Mic19 loss leads to decreased mitochondria-ER contact, disorganization of mitochondrial cristae junctions, decreased levels of mitochondrial ETC, and indicators of lowered FAO. These all correspond to a striking in vivo NASH phenotype. The authors propose that Mic19 acts to transport phosphatidic acid (PA) into mitochondria where it is needed for cardiolipin synthesis (CL), and that the phenotypes of Mic19 loss result from loss of this function. This is an impressive amount of work, and a beautiful phenotype. Unfortunately, the data are over interpreted and while suggestive, do not test the causality of the mechanism proposed by the authors.

Major points:

The only rescue experiment in the paper is re-expression of Mic19 in LKO mice. Thus, the only conclusion we can draw is that all the observed phenotypes, from molecular differences in phospholipid abundance, to overt fat accumulation in LKO livers, result from Mic19 loss. The phenotypes are correlated, but we cannot begin to put them into a pathway and say that one leads to another, yet the authors make these types of claims over and over again throughout the manuscript (e.g. in the abstract: “impairment of ER-mitochondria contacts by Mic19... ..causes reduction of CL” and “reduction of ER-mitochondrial contacts and disorganization of mitochondrial cristae result in mitochondrial unfolded protein stress response (UPRmt) in mouse hepatocytes, impairing liver mitochondrial fatty acid β -oxidation and lipid metabolism, which further spontaneously triggers nonalcoholic steatohepatitis (NASH)”). These unproven statements occur in every section of the paper. In the absence of further mechanistic experiments, the language in the manuscript must be edited to make clear that a causal relationship has not been established, other than that Mic19 loss results in each of these changes.

We greatly appreciate the reviewer’s comments and suggestions. According to the reviewer’s comments, we performed liposome extraction and phospholipid transfer assay in vitro, separately. The data reveal that Mic19 or SC25A46 are able to bind to and transfer PA in vitro (Figures 4F, 4G, 4J, 4K, S4E and S4F of the revised manuscript). Therefore, the impairment of ER-mitochondria contacts by Mic19 depletion could reduce PA transport from ER to mitochondria, then decreasing CL production since mitochondrial PA is a precursor of CL synthesis. Additionally, according to the reviewer’s suggestions, we edited the language related

to some statements in the revised manuscript.

In particular, the authors purport that all the downstream effects they see of Mic19 knockout (e.g. membrane disorganization, ETC complexes, FAO, NASH etc) are a result of blocked PA transport into mitochondria and resulting decreased cardiolipin synthesis. However, the authors cannot conclude this from the data. Mic19 LKO mice show decreases in every phospholipid class in their mitochondria (Figure 3A-B), likely due to loss of MICOS as a whole (Figure 5A). Loss of MICOS likely contributes to membrane disorganization completely in the absence of changes in membrane lipid composition. While other mechanistic studies have linked CL to ETC function and mitoUPR in the past, there are no experiments herein that show the specificity of the CL effect on the downstream phenotypes in this study. To make these claims, the authors need to devise a way to specifically rescue CL vs. other mitochondrial lipids in this model. Again, in the absence of mechanistic experiments, which could easily be performed in their cell line systems, the language in the manuscript must be edited to make clear that a causal relationship between decreased mitochondrial CL and the other measured phenotypes has not been established.

We agree to the reviewer's comments that the downstream effects after Mic19 knockout (e.g. membrane disorganization, ETC complexes, FAO, NASH etc) are complicated in mouse liver in vivo. Besides impairing of PA transport, Mic19 KO also causes the loss of MICOS complex that is also associated with the metabolism or transport of some other phospholipids including CL, PS (Aaltonen et al., 2016; Anand et al., 2020). We also overexpressed CLS1 (CL synthesis enzyme) to rescue CL in Mic19 KO cells, but CL was not rescued (data not shown). It may be that Mic19 KO impairs PA transport, resulting in the loss of CL synthesis precursor (PA is a CL synthesis precursor), but did not impair the activity of CL synthesis.

Additionally, our data show that Mic19 depletion reduces CL production (Figure 3 of the revised manuscript), which is associated with mitochondrial dysfunction including the weakened mitochondrial respiration, reduced ATP production and increased ROS levels, etc. (Figures 1, 3, 4, 5 and S5 of the revised manuscript). These data are consistent with the previous report that CL remodeling links oxidative stress and mitochondrial dysfunction (Li et al., 2010).

Mic19 did not interact with PC, PS and PE, but PC, PS and PE were also reduced in mitochondria of Mic19 KO cells (Figure 3B of the revised manuscript). It is probably that Mic19 depletion led to a reduction in the ER-mitochondria contacts and accompanied by an increased distance between these two organelles (Figure 1 of the revised manuscript). Since PC and PS can also transport from ER to mitochondria, and the Mic19 KO-caused the reduction of

ER-mitochondria contacts may impair the transport of PC and PS to mitochondria. In addition, PS is the precursor for PE, the decreased PE may be the consequence of the reduced PS. Similarly, previous studies have reported that Mfn2 knockout also causes a reduction of PS, PC and PE in liver mitochondria although Mfn2 just binds to PS (Hernández-Alvarez et al., 2019). We have discussed this issue in the revised manuscript.

In addition, according to the reviewer's suggestion, we edited the related language, modified some conclusions and discussed the related issues in the revised manuscript.

References

Aaltonen, M.J., Friedman, J.R., Osman, C., Salin, B., di Rago, J.P., Nunnari, J., Langer, T., and Tatsuta, T. (2016). MICOS and phospholipid transfer by Ups2-Mdm35 organize membrane lipid synthesis in mitochondria. *The Journal of cell biology* 213, 525-534.

Anand, R., Kondadi, A.K., Meisterknecht, J., Golombek, M., Nortmann, O., Riedel, J., Peifer-Weiß, L., Brocke-Ahmadinejad, N., Schlütermann, D., Stork, B., et al. (2020). MIC26 and MIC27 cooperate to regulate cardiolipin levels and the landscape of OXPHOS complexes. *Life science alliance* 3. e202000711

Li, J., Romestaing, C., Han, X., Li, Y., Hao, X., Wu, Y., Sun, C., Liu, X., Jefferson, L.S., Xiong, J., et al. (2010). Cardiolipin remodeling by ALCAT1 links oxidative stress and mitochondrial dysfunction to obesity. *Cell metabolism* 12, 154-165.

Hernández-Alvarez, M.I., Sebastián, D., Vives, S., Ivanova, S., Bartoccioni, P., Kakimoto, P., Plana, N., Veiga, S.R., Hernández, V., Vasconcelos, N., et al. (2019). Deficient Endoplasmic Reticulum-Mitochondrial Phosphatidylserine Transfer Causes Liver Disease. *Cell* 177, 881-895.e817.

The authors draw conclusions about fatty acid synthesis and oxidation from surrogate measures that do not directly assay these processes. In Figure 6, ACC1 expression is definitely increased in LKO mice. However, the authors only discuss that the pACC1/ACC1 ratio is not changed. I do not agree with the interpretation that this means fatty acid synthesis can be ruled out as unaffected. Similarly, changes in fatty acid oxidation are inferred from differences in gene expression that are only detectable in fasted mice. The authors need to provide functional assays for fatty acid synthesis and oxidation, or revise the language to be more suggestive and less conclusive.

We appreciate the reviewer's comments and suggestions. In the de novo lipogenesis pathway, acetyl-CoA is first carboxylated to form malonyl-CoA, which then followed by a series of oxidation reactions to form triglycerides. This is followed by a series of oxidation reactions that ultimately result in the production of triglycerides. The key enzymes involved in this process are Acc1, p-Acc1, and Fasn. Acc1 participates in the synthesis of malonyl-CoA, while p-Acc1 inhibits the activity of Acc1. Although both Acc1 and p-Acc1 in the liver of Mic19 LKO mice were upregulated, the ratio between the two enzymes showed no significant difference (Figures S6G and S6H of the revised manuscript), indicating that the activity of

whole *Acc1* is not increased in *Mic19* LKO mice. Additionally, we measured the content of malonyl-CoA in liver tissue from control and *Mic19* LKO mice, and there is no difference between the two groups (Figure S6E of the revised manuscript). Therefore, we speculated that the synthesis of fatty acid in *Mic19* LKO mice did not increase. Moreover, we found that the mRNA levels of lipid synthesis and fatty acid chain elongation related genes (*Mlycd*, *Fasn*, *Gpat2*, *Agpat1*, *Lpin1*, *Dgat2*) were not changed in *Mic19* LKO mouse liver compare to control mouse liver (Figure S6F of the revised manuscript). In addition, it is well known that mitochondrial fatty acid β -oxidation can lead to ketone body production (ketogenesis) by the liver under fasted conditions (Lee et al., 2017). We found that the ketogenesis product of β -hydroxybutyrate (β HB, one of ketone bodies) was remarkably reduced in *Mic19* LKO mice under both fed and fasted conditions (Figure 6K of the revised manuscript), and the mRNA levels of ketogenesis related genes *Hmgcs2* and *Hmgcl* were decreased in *Mic19* LKO mouse liver (Figure 6L of the revised manuscript), indicating that mitochondrial fatty acid β -oxidation is impaired in *Mic19* LKO mouse liver.

It should be noted that, as the reviewers pointed out, we did not perform a direct functional analysis of fatty acid synthesis and oxidation. Therefore, we can only conclude that *Mic19* LKO mice exhibit abnormalities in lipid metabolism, which is most likely due to the decreased lipolysis, and finally leading to fat accumulation in mouse liver. As suggested by the reviewer, we edited the language to be suggestive and less conclusive in the revised manuscript.

Reference

Lee, J., Choi, J., Selen Alpergin, E.S., Zhao, L., Hartung, T., Scafidi, S., Riddle, R.C., and Wolfgang, M.J. (2017). Loss of Hepatic Mitochondrial Long-Chain Fatty Acid Oxidation Confers Resistance to Diet-Induced Obesity and Glucose Intolerance. *Cell reports* 20, 655-667.

*How *Mic19* is connected to changes in FAO gene expression is unclear. Do the authors propose it is through mitoUPR and/or ER stress signaling? If so, they need to test those claims using inhibition or genetic manipulation of those pathways to show their causal involvement, or make clear that this is a hypothesis.*

We appreciate the reviewer's comments and suggestions. To verify the relationship between ER stress and reduced fatty acid oxidation, *Mic19* LKO mice were administered intraperitoneal injection of the ER stress inhibitor TUDCA (tauroursodeoxycholate) (100 mg/kg/day for consecutive 14 days) (Wang et al., 2022). Then the expression of genes related to fatty acid oxidation were analyzed by qRT-PCR. TUDCA significantly inhibited the ER stress in *Mic19* LKO mice, and upregulated the expression of genes related to β -oxidation (Figure S6J of the revised manuscript), indicating that *Mic19* LKO could cause ER stress, which is also contribute

to the downregulation of FAO gene expressions.

Reference

Wang, Q., Zhou, H., Bu, Q., Wei, S., Li, L., Zhou, J., Zhou, S., Su, W., Liu, M., Liu, Z., et al. (2022). Role of XBP1 in regulating the progression of non-alcoholic steatohepatitis. *Journal of hepatology* 77, 312-325.

In Figure 9, the authors show correlative changes in Mic19 and Mic60 expression in a model of NASH. There is nothing in this figure to suggest these changes are causal. The text must reflect this, or alternatively, the authors could perform rescue experiments with their Mic19 AAV in this model.

As suggested by the reviewer, we investigated the effect of Mic19 overexpression on MCD-induced liver disease. adeno-associated virus encoding Mic19-Flag were injected by a tail vein into 8-week-old MCD mice. After overexpression of Mic19-Flag in MCD-treated mouse liver, the upregulation of TG level and mRNA level of inflammation-related genes *Cd68*, *Cxcl10* and *Tnf* were suppressed and recovered to be similar to that of normal diet mice (Figures S9G-S9I of the revised manuscript). Moreover, H&E staining revealed that Mic19-Flag overexpression remarkably suppressed MCD-caused hepatic steatosis in mice (Figure S9J of the revised manuscript). Therefore, Mic19 overexpression suppresses MCD-induced fatty liver disease. The new data were provided in the revised manuscript.

Minor points:

For all the mito-ER colocalization studies, what is the total change in mitochondrial surface area in the knockouts/knockdowns? It appears by eye that MICOS loss decreases mitochondrial abundance. If this is true, how is it taken into account? If there are fewer mitochondria, then won't there naturally be less ER-mitochondria contact?

We greatly appreciate the reviewer's comments. We investigated the effect on Mic19 depletion on mitochondrial number and content, which may impair ER-mitochondria contacts. HIS-SIM imaging and Western blotting analysis showed that mitochondrial number and contents (mitochondrial marker proteins including Tom20, Tom40, Tim23, SDHA and Cox4) was not changed in Mic19 KO cells (Figures S1E-S1G of the revised manuscript). Additionally, according to the reviewer's suggestion, we normalized "the number of ER-mito contacts" to "10 μm mitochondrial membrane surface", and provided the new statistical data in Figures 1I, 2M of the revised manuscript.

In Figure 2, it appears that GAPDH is increased in the western blots, which exaggerates the degree to which SLC25A46 and Mic19 affect one another's protein levels. The language that is used to describe this in the text exaggerates this finding.

We sincerely thank the reviewer's comments. Actually, we calculated the protein level of SLC25A46 or Mic19 to the protein level of GAPDH to get the "the relative protein level" in the original manuscript. According to the reviewer's suggestion, we repeated Western blotting analysis and densitometry analysis, the new data were provided in the Figures 2C-2F of the revised manuscript. In addition, we also edited the language to re-describe this finding in the text of the revised manuscript.

It would be helpful to have Figure S3A earlier, as the authors perform experiments in LKO mice prior to this figure.

According to the reviewer's suggestion, we have displayed "Figure S3A" earlier and prior to the experiments in LKO mice, but we added a new Figure S2 in the revised manuscript, so the number is still "Figure S3A" in the revised manuscript.

Figure 5C is uninterpretable. The image is too poor quality to assess their conclusions, it is not possible to see cristae in the fl/fl control.

We sincerely thank the reviewer for his/her comments. According to the reviewer's comments, we have repeated the electron microscope analysis of mouse liver tissue, and the new images were provided in Figure 5C of the revised manuscript.

Figure S4I is unnecessary, this is a standard technique and it is unclear why the authors felt it was needed to have a schematic (as opposed to any other experiment in the paper).

As suggested by the reviewer, we have deleted the schematic representation of the RNA-seq experiment in the revised manuscript.

The authors show convincing data indicating the accumulation of liver and serum FFA, and liver TG in LKO mice. They find decreases in some FAO transcripts, and conclude that this results in decreased fatty acid oxidation in LKO mice. They also show convincing decreases in ETC abundance in LKO mice. However, Figure S4 shows increased oxygen consumption and CO2 production. Why? This is not well explained or connected to the rest of the data in the paper. Are the elevated serum FFA driving increased FAO and respiration in other tissues like muscle?

We appreciate the reviewer's comments and suggestions. Mic19 LKO not only led to NASH and liver fibrosis, but also caused less body weight and lean phenotype (Figures 7, S7 and S8 of the revised manuscript), indicating that Mic19 may also impair the function of other organs in mice. Mic19 LKO mice displayed remarkably increased oxygen consumption (VO₂) and carbon dioxide production (VCO₂) rates by mice metabolism cage experiment (Figures S6K-S6N of the revised manuscript). As adipose tissue is an important organ in regulating energy balance, serving not only as an energy store but also as a modulator of metabolism. We found that the weight of adipose tissues including eWAT (epididymal white adipose tissue) and iWAT (inguinal white adipose tissue) were slightly decreased (Figure S6O of the revised manuscript). However, qRT-PCR analysis revealed that the mRNA levels of some mitochondrial β -oxidation genes including Cpt1b, Cpt2 and Acads were significantly increased in Mic19 LKO mouse iWAT (Figure S6P of the revised manuscript). These results indicate that lipolysis is increased in the iWAT of Mic19 LKO mice, which may be a compensatory response of the impairment of β -oxidation in the liver tissue, thus increasing the energy expenditure in Mic19 LKO mice. In addition, Mic19 LKO-increased lipolysis in the iWAT may cause lean phenotype and the increased plasma FFA level in Mic19 LKO mice. We have provided these data and discussed the issue in the revised manuscript.

Reviewer #4

In this paper, the authors examined the function of the MICOS subunit Mic19, at the molecular, cellular and physiological levels. The first part of the study focuses on the role of Mic19 and its partners, including SLC25A46, in ER-mitochondria contact regulation, transport and metabolism of mitochondrial lipids, such as PA, and protein-protein and protein-lipid interactions. For this, the authors used a variety of techniques ranging from super resolution microscopy and EM imaging, pull down assays from Hela control, Mic19 KO or shMic19, and shSLC25A46 cells. Lipidomics data from hepatic mitochondrial fractions of Mic19 liver-specific KO mice are also presented and biochemical assays with purified proteins are shown. The second part of the paper focuses on the pathophysiological consequences related to the absence of Mic19 in the liver. More specifically, the paper reports that Mic19 is required for mitochondrial membrane organization, and its absence leads to a number of mitochondrial defects (UPR(mt), oxidative phosphorylation impairment, decreased ATP production, increased ROS levels) and ER stress. Linked to this, Mic19 LKO mice show liver FA metabolism defects due to FA beta-oxidation. This leads to increased TG levels and hepatic steatosis, as well as liver inflammation and finally liver fibrosis. Of note, the authors have performed rescue experiments by re-expressing a FLAG-tagged version of Mic19, which reverse the phenotype described. Overall, the absence of Mic19 triggers liver disease (NASH, fibrosis) via mitochondria function impairment. In the last section of the manuscript, the authors trigger the liver disease (NASH) by a specific diet, and show that consequently Mic19 expression decreased in the liver, whereas it was shown before that Mic19 depletion was the cause of liver disease. The second part of the paper is well executed and is impressive in terms of metabolic data collected, although I don't know what conclusion to draw from the MCD-induced fatty liver disease model, as Mic19 is certainly not the only protein whose expression level varies following such treatment. The paper is generally well written. The data are well put together and follow a logical order. My main concerns are with the first part of the paper and final conclusions about the mechanism by which Mic19 supposedly operates.

Major points

1. I'm not convinced by the decrease in colocalization between ER and mitochondria as presented in the figures 1A, 1B, 1C, 1D, 2F, 2G and quantification shows very modest effects at best. There is still a lot of MAMs. Therefore, the authors should moderate their claims: the regulatory effect on contact sites formation by SLC25A46 or Mic19 is minor. This suggests that the effects obtained latter in vivo are probably not due to a defect in the formation of contact

sites between ER and mitochondria. By contrast, mitochondria are in a very bad state, and shape, in the absence of Mic19 compared to controls, this is obvious in fig1E. The differences in morphology (shorter, round) should be addressed with more emphasis. Are there less mitochondria in Mic19 LKO cells? Is there mitophagy occurring?

We greatly appreciate the reviewer's comments and suggestions. We agree to the reviewer's opinion that the effects obtained by Mic19 LKO in mice in vivo are complicated and are probably due to many factors including ER-mitochondria contacts, mitochondrial morphology, number and structure. According to the reviewer's comments, we further examined mitochondrial number, length, structure and mitophagy of control and Mic19 KO cells. We found that mitochondrial number, length, contents and mitophagy were not changed in Mic19 KO cells (Figures 1C, 1E, 1F, 5C, S1E-S1H of the revised manuscript), but mitochondrial cristae morphology was remarkably changed and mitochondrial cristae junctions were dramatically decreased in Mic19 KO cells or Mic19 LKO liver cells (Figures 1E, 1G, 5F, 5G of revised manuscript), which may cause more round mitochondria in Mic19 LKO liver cells (Figure 5C of the revised manuscript). It should be noted that ER-mitochondria contacts are highly associated with mitochondrial structure since ER-mitochondria contacts provide major phospholipids including PA for the biogenesis of mitochondrial cristae membrane.

According to the reviewer's suggestions, we pointed out that mitochondrial structure may also highly contribute to the effect of Mic19 LKO in vivo and discussed the issue in the revised manuscript.

2. Lipidomics. First, it is unclear based on the data provided whether the authors have properly purified mitochondria from MAMs, ER and others organelles. No western blot showing the evidence is provided. Second, the problem is that not only PA or CL decreased in Mic19 LKO mitochondria, but also all analyzed lipids, including PC, PE and PS, which questions the specificity of the EMC2-SLC25A46-Mic19 axis with respect to PA (although this is the proposed model in fig 4F and S8).

Related to this, I disagree with the conclusion from figure 3 (ie. last sentence from the first paragraph page 5.) Since all lipids analyzed decreased, it is possibly not only a problem of CL synthesis. Moreover, no experiment showed that the trafficking of PA from the ER to mitochondria was affected. Mic19 is a subunit of MICOS at the OMM-IMM interface, not acting as a PA transporter between ER and mitochondria.

We greatly appreciate the reviewer's comments and suggestions. According to the reviewer's suggestion, we analyzed the purity of the mitochondrial fraction used for lipidomics analysis

by Western blotting. Western blotting analysis showed mitochondrial marker Tom40 existed only in the fraction of mitochondria but not ER or MAM (mitochondria-associated membrane) (Figure S3C of the revised manuscript), indicating that mitochondrial fraction analyzed for lipidomics was pure.

In addition, our results indicate that Mic19 did not interact with PC, PS and PE, but PC, PS and PE were also reduced in mitochondria of Mic19 KO cells (Figure 3B of the revised manuscript). It is probably that Mic19 depletion led to a reduction in the ER-mitochondria contacts and accompanied by an increased distance between these two organelles (Figure 1 of the revised manuscript). Since PC and PS can also transport from ER to mitochondria, and the Mic19 KO-caused the reduction of ER-mitochondria contacts may impair the transport of PC and PS to mitochondria. In addition, PS is the precursor for PE, the decreased PE may be the consequence of the reduced PS. We have discussed this issue in the revised manuscript.

Also, Mic19 deletion results in a reduction of the MICOS complex subunits, including Mic60 (Figures 5A and 5B of the revised manuscript), Mic26 and Mic27 (Guarani et al., 2015). Mic60 could co-sediment with liposomes and regulates membrane shaping (Zhou et al., 2022), and Mic26 or Mic27 (the subunit of MICOS complex) are also associated with cardiolipin metabolism (Anand et al., 2020). Moreover, the alterations of inner membrane morphology may be linked to membrane lipid metabolism and synthesis (Tasseva et al., 2013), thus, the alterations of inner membrane morphology may potentially influence the localization of enzymes and proteins on the mitochondrial inner membrane and impair phospholipid synthesis. Therefore, Mic19 may directly or indirectly regulate lipid metabolism and transport. In this manuscript, we focus on the role of Mic19 on PA binding and transport. We have discussed the issues in the revised manuscript.

To further investigate the role of Mic19 in PA transport, we performed phospholipid transfer assay *in vitro*. phospholipid transfer assay indicates that Mic19 or SLC25A46 was able to participate in PA transfer directly *in vitro* (Figures 4G, 4K and S4F of the revised manuscript). In addition, we previously reported that Mic19 locates in mitochondrial inner membrane space (Tang et al., 2020), therefore, Mic19 may participate in PA transport from OMM to IMM. We have discussed the issue in the revised manuscript.

Additionally, according to the reviewer's comments, we edited some conclusions (including last sentence from the first paragraph page 5 in the original manuscript) in the revised manuscript.

References

- Anand, R., Kondadi, A.K., Meisterknecht, J., Golombek, M., Nortmann, O., Riedel, J., Peifer-Weiß, L., Brocke-Ahmadinejad, N., Schlütermann, D., Stork, B., et al. (2020). MIC26 and MIC27 cooperate to regulate cardiolipin levels and the landscape of OXPHOS complexes. *Life science alliance* 3, e202000711.
- Guarani, V., McNeill, E.M., Paulo, J.A., Huttlin, E.L., Fröhlich, F., Gygi, S.P., Van Vactor, D., and Harper, J.W. (2015). QIL1 is a novel mitochondrial protein required for MICOS complex stability and cristae morphology. *eLife* 4, e06265
- Tasseva, G., Bai, H.D., Davidescu, M., Haromy, A., Michelakis, E., and Vance, J.E. (2013). Phosphatidylethanolamine deficiency in Mammalian mitochondria impairs oxidative phosphorylation and alters mitochondrial morphology. *The Journal of biological chemistry* 288, 4158-4173.
- Zhou, J., Duan, M., Wang, X., Zhang, F., Zhou, H., Ma, T., Yin, Q., Zhang, J., Tian, F., Wang, G., et al. (2022). A feedback loop engaging propionate catabolism intermediates controls mitochondrial morphology. *Nature cell biology* 24, 526-537.
- Tang, J., Zhang, K., Dong, J., Yan, C., Hu, C., Ji, H., Chen, L., Chen, S., Zhao, H., and Song, Z. (2020). Sam50-Mic19-Mic60 axis determines mitochondrial cristae architecture by mediating mitochondrial outer and inner membrane contact. *Cell death and differentiation* 27, 146-160.

3. NAO. NAO is used at 10 μ M: it is known that at high concentration NAO fluorescence is quenched (PMID 12124423) especially in zones where NAO is confined such as in mitochondria. The green fluorescence of NAO may decrease in the presence of CL (PMID 24372165, 1396703) and could also depend on membrane potential. There is no control from the authors that the drop of NAO fluorescence is due to CL decrease or due to something else (self-quenching, membrane potential). Given that NAO is used at high concentration (10 μ M), data should be interpreted with caution. The authors must perform these controls.

We appreciate the reviewer's comments and suggestions. At high concentration NAO fluorescence is quenched, and loading and retention of NAO is dependent upon membrane potential in living cells (Jacobson et al., 2002). Thus, we stained live cells with NAO at the concentration of 100 nM (not 10 μ M) in the original manuscript. Additionally, we detected the mitochondrial membrane potential in Mic19 KO and shSLC25A46 cells by TMRM staining. Mic19 KO nor SLC25A46 knockdown (KD) led to a decrease in mitochondrial membrane potential (Figures S3H and S3I of the revised manuscript), suggesting that the decrease of mitochondrial membrane potential may impair the green fluorescence of NAO in living cells. Therefore, we fixed the cells and then performed NAO staining (not depend on mitochondrial membrane potential), and followed by fluorescence analysis using flow cytometry; flow cytometry analysis showed that the NAO fluorescence intensity (indicate cardiolipin level) was decreased in Mic19 KO or SLC25A46 KD cells (Figures 3G-3J of the revised manuscript). Moreover, to further confirmed the effect of Mic19 or SLC25A46 depletion on the level of cardiolipin, we performed lipidomics analysis. The CL level was decreased in the mitochondria

fractions of Mic19 KO or SLC25A46 KD cells (Figures 3C-3F of the revised manuscript).

Together, NAO staining and lipidomics analysis data suggest that Mic19 KO and SLC25A46 KD decrease the level of CL in mitochondria. We have provided the new data in the revised manuscript.

Reference

Jacobson, J., Duchen, M.R., and Heales, S.J. (2002). Intracellular distribution of the fluorescent dye nonyl acridine orange responds to the mitochondrial membrane potential: implications for assays of cardiolipin and mitochondrial mass. *Journal of neurochemistry* 82, 224-233.

4. Biochemical assays (figure 4). There is no evidence that Mic19-His and SLC24A46-His are correctly purified based on the WB. Standards and SDS-PAGE images must be shown, and a more detailed protocol in Method section must be provided.

According to the reviewer's suggestions, we have performed the SDS-PAGE assay for purified Mic19-His and SLC25A46-His, SDS-PAGE and Coomassie Blue staining images showed Mic19-His and SLC25A46-His are purified (Figures 4A, 4B, 4H and 4I of the revised manuscript). Additionally, we have added the protein purification methods in the section of "Materials and Method" in the "Supplementary information" of the revised manuscript.

Protein-lipid overlay assays. Lipids in such arrays are not presented to the proteins as they should be, that is, in a lipid bilayer context. Lipid concentrations in spots are also much higher than in cell situation. No reliable conclusion can be made based on protein-lipid overlay assay. They may serve as pilot experiment at best.

We appreciate the reviewer's comments. To validate the binding of Mic19 and SLC25A46 with phosphatidic acid, we performed liposome extraction experiments (Figures 4F, 4J and S4E of the revised manuscript). The results revealed that both Mic19 and SLC25A46 could successfully extract PA. In addition, we performed liposome floatation experiments, which showed that Mic19 could bind to PA (Figures 4E and S4D of the revised manuscript). Furthermore, we performed phospholipid transfer assay in vitro. Phospholipid transfer assay indicates that Mic19 or SLC25A46 was able to participate in PA transfer directly in vitro (Figures 4G, 4K and S4F of the revised manuscript).

Liposome sedimentation. There is little information of how the experiment was made : concentration of proteins and lipids used? Are liposomes made of pure lipids (PA or PC or PE or PS) (as indicated in fig 4D)? No image of SDS-PAGE is provided, but a WB, no quantification is shown. The experiment has been repeated?

In the liposome co-precipitation assay, 10 μ l of protein (about 10 μ g) were incubated with 45 μ l of liposomes (2 mM). The liposomes are not composed of pure lipids but of a mixture of 80% E. coli lipid extract and 20% PC, or PS, or PA, or PE. Additionally, we have provided the SDS-PAGE and Coomassie Blue staining images for Mic19 and SLC25A46 in the revised manuscript (Figures 4A, 4B, 4H and 4I of the revised manuscript). The liposome co-sedimentation assay was performed for three times. Additionally, we have added “Liposome co-sedimentation assay” in the section of “Materials and Method” in the “Supplementary information” of the revised manuscript.

SLC25A46 is a transmembrane protein and therefore cannot be used in lipid binding assays as with a soluble protein. I also have doubts about the authors' method of purification of SLC25A46. Does SLC25A46 get inserted into proteoliposomes?

We appreciate the reviewer’s comments. We performed the SDS-PAGE assay for purified SLC25A46-His, SDS-PAGE and Coomassie Blue staining images showed SLC25A46-His are purified (Figures 4H and 4I of the revised manuscript). In addition, we analyzed the SLC25A46 protein structure by using DisoLipPred and InterPro (2 online tools) predictions. The results show that beside to contain transmembrane region, SLC25A46 protein also contains some disordered lipid binding regions (Figures S4A of the revised manuscript), indicating that SLC25A46 may bind to lipid by its lipid binding regions. It should be noted that SLC25A46 may not be totally located at mitochondrial outer membrane, part of SLC25A46 protein may be located outside of mitochondrial outer membrane and bind to lipid. In our future study, we will investigate which part of SLC25A46 bind to lipid.

Additionally, we performed phospholipid transfer assay in vitro. Phospholipid transfer assay indicates that Mic19 or SLC25A46 was able to participate in PA transfer directly in vitro (Figures 4G, 4K and S4F of the revised manuscript).

The model shown in Figure 4F is not scientifically supported. The authors did not present any PA transfer data, but merely a crude PA binding assay that should be interpreted with caution, even concern for SLC25A46. Therefore, the claim “Together, the EMC2-SLC25A46-Mic19 axis regulates ER-mitochondria contacts and participates in ER-mitochondrial PA transport” (First sentence in page 6) is not supported by the data: the effects on ER-mitochondria contacts are minor and PA transport is not analyzed.

According to the reviewer’s comments, we performed phospholipid transfer assay in vitro. phospholipid transfer assay indicates that Mic19 or SLC25A46 was able to participate in PA transfer directly in vitro (Figures 4G, 4K and S4F of the revised manuscript).

5. Link between ER-mitochondria contact/PA transport and metabolic defects. Page 7: “Taken together, Mic19 LKO-caused reduction of ER-mitochondria contacts leads to UPRmt and ER stress”. Discussion: the “impairment of PA transport is involved in the development and progression of liver diseases”. These bold claims describe a correlation that has been made experimentally by the authors, but are not demonstrated as one causing the other. MICOS is one of the major structural elements of mitochondria that participate in many processes, such as generation of mitochondrial architecture, protein import into mitochondria, and lipid metabolism. Therefore, destabilization of MICOS by the absence of Mic19 may be the source of other problems than lipid transfer, suggesting that the metabolic issues observed here are possibly not related to PA transfer.

We greatly thank the reviewer’s comments. Mic19 deletion results in a reduction of the MICOS complex subunits, including Mic60 (Figures 5A and 5B of the revised manuscript), Mic26 and Mic27 (Guarani et al., 2015). Mic60 could co-sediment with liposomes and regulates membrane shaping (Zhou et al., 2022), and Mic26 or Mic27 (the subunit of MICOS complex) are also associated with cardiolipin metabolism (Anand et al., 2020). Moreover, the alterations of inner membrane morphology may be linked to membrane lipid metabolism and synthesis (Tasseva et al., 2013), thus, the alterations of inner membrane morphology may potentially influence the localization of enzymes and proteins on the mitochondrial inner membrane and impair phospholipid synthesis. Therefore, Mic19 may directly or indirectly regulate lipid metabolism and transport, causing the metabolic defects in mouse liver. In addition, we admit that the mechanism of metabolic defects in Mic19 LKO mice is complicated and may be the results of multiple factors, and our data indicate that the PA transport is highly related. In this manuscript, we focus on the physiological role of Mic19 on PA binding and transport. We have discussed the issues and modified some conclusions in the revised manuscript.

References

- Anand, R., Kondadi, A.K., Meisterknecht, J., Golombek, M., Nortmann, O., Riedel, J., Peifer-Weiß, L., Brocke-Ahmadinejad, N., Schlütermann, D., Stork, B., et al. (2020). MIC26 and MIC27 cooperate to regulate cardiolipin levels and the landscape of OXPHOS complexes. *Life science alliance* 3, e202000711.
- Guarani, V., McNeill, E.M., Paulo, J.A., Huttlin, E.L., Fröhlich, F., Gygi, S.P., Van Vactor, D., and Harper, J.W. (2015). QIL1 is a novel mitochondrial protein required for MICOS complex stability and cristae morphology. *eLife* 4, e06265
- Tasseva, G., Bai, H.D., Davidescu, M., Haromy, A., Michelakis, E., and Vance, J.E. (2013). Phosphatidylethanolamine deficiency in Mammalian mitochondria impairs oxidative phosphorylation and alters mitochondrial morphology. *The Journal of biological chemistry* 288, 4158-4173.
- Zhou, J., Duan, M., Wang, X., Zhang, F., Zhou, H., Ma, T., Yin, Q., Zhang, J., Tian, F., Wang, G., et al. (2022). A feedback loop engaging propionate catabolism intermediates controls mitochondrial morphology. *Nature cell biology* 24, 526-537.

6. Rationale of figure 9. The authors treated mice to induce liver disease. They observed that the expression level of Mic19 decreases (as perhaps dozens of other proteins?) and concluded that their results are consistent with the function of Mic19 in repressing mouse liver disease. There is a problem of cause and consequence here that the authors do not explain.

According to the reviewer's comments, we investigated the effect of Mic19 overexpression on MCD-induced liver disease. Adeno-associated virus encoding Mic19-Flag were injected by a tail vein into 8-week-old MCD mice. After overexpression of Mic19-Flag in MCD-treated mouse liver, the upregulation of TG level and mRNA level of inflammation-related genes *Cd68*, *Cxcl10* and *Tnf* were suppressed and recovered to be similar to that of normal diet mice (Figures S9G-S9I). Moreover, H&E staining revealed that Mic19-Flag overexpression remarkably suppressed MCD-caused hepatic steatosis in mice (Figure S9J). These data suggest that re-expression of Mic19 in mice could alleviate MCD diet-induced liver diseases, indicating that MCD diet -caused reduction of Mic19 level may be contributed to MCD diet-induced liver diseases.

Other points

a. Page 2: "how PA transport between ER and mitochondria is still unknown". I assume the authors meant "how PA is transported...". In fact there are several recent papers that describe mechanisms driven by lipid-transfer proteins to transport PA between the ER and mitochondria; they should be cited here (For example PMID 33938112, 36693319, 30093493, 36282247).

We sincerely thank the reviewer's comments. We have cited these papers in the revised manuscript.

b. Is there any evidence that EMC2 and SLC25A46 promote MAM formation through direct physical interactions? In the introduction, the authors state this as a fact, although I'm not sure whether this is actually described. In the paper (ref#36) cited by the authors, EMC2 was found among many others proteins in a bioID screen, which is not interaction per se, but proximity. See also the first sentence of second paragraph in page 4.

We appreciate the reviewer's comments. Janer *et al* reported that SLC25A46 could interact with EMC2 by the BioID assay (Janer et al., 2016). In addition, according to the reviewer's comments, we further performed co-IP assay and showed that the exogenous expressed and endogenous SLC25A46 could interact with EMC2 (Figures 2A and 2B of the revised manuscript). Moreover, we found that SLC25A46 regulates ER-mitochondria contacts (Figures 2G-2M of the revised manuscript). Therefore, SLC25A46-EMC2 interaction may promote MAM formation. We have changed some descriptions and conclusions related to SLC25A46-EMC2 interaction in the revised manuscript.

Reference

Janer, A., Prudent, J., Paupe, V., Fahiminiya, S., Majewski, J., Sgarioto, N., Des Rosiers, C., Forest, A., Lin, Z.Y., Gingras, A.C., et al. (2016). SLC25A46 is required for mitochondrial lipid homeostasis and cristae maintenance and is responsible for Leigh syndrome. *EMBO molecular medicine* 8,1019-1038.

c. In figure 3B, data are presented as fold change versus control mice. Then why values for lipids from control mice (Mic19^{flox/flox}) are in red color in the heatmap (which suggest more lipids compared to control)?

We greatly appreciate the reviewer's comments and apologize for the mistake in the description of the original manuscript. In the Figure 3B, we firstly measured the relative contents of different phospholipids from Mic19^{flox/flox} and LKO mice by lipidomics. Then, according to the relative level of the phospholipids, Z-score, which is always used to reflect the relative level of metabolites, was converted and was used for heatmap analysis. When the levels of metabolites were lower than average level of the whole samples, the Z-score in this sample was shown as negative. Therefore, smaller Z-score means lower level of targeted metabolites. In order to show the relative abundance of phospholipids, the heatmap was shown as blue (negative Z-score) and red (positive Z-score) colors. As shown in the chart, redder colors indicated higher expression and bluer colors indicated lower expression. The figure legend of Figure 3B has been changed in the revised manuscript.

d. in vivo experiments from fig5A, B are convincing, but TEM images in fig 5C are not clear and of poor quality. What are we supposed to see? Authors should identify organelles and contact between them. Cristea number are quantified in fig 5E but no cristae are seen in fl/fl condition from fig5C.

We sincerely thank the reviewer for his/her comments. According to the reviewer's comments, we have repeated the transmission electron microscope (TEM) analysis of mouse liver tissue and provided the new images (Figure 5C of the revised manuscript). The red arrows showed the ER-mitochondrial contacts.

e. The sentence "the levels of liver and serum free fatty acids were dramatically elevated in Mic19 LKO mice (Figures 6F and 6G)." is written in the text in page 7 and again in page 8.

According to the reviewer's suggestion, we have deleted the repeated sentence "the levels of liver and serum free fatty acids were dramatically elevated in Mic19 LKO mice (Figures 6F and 6G)." in the revised manuscript.

Reviewers' Comments:

Reviewer #1:

Remarks to the Author:

The authors addressed my concerns. I suggest moving Figure S9G-J to main Figure 9.

Reviewer #2:

Remarks to the Author:

The authors have performed some additional experiments of which some are convincing, yet the major concern has not been resolved sufficiently in my opinion. The claim about the role of MIC19 in PA transport is grossly overstated. As we previously said, we still have to say that there is no sufficient "...biochemical evidence that Mic19 or SLC25A46 directly mediate the transport of PA, or between which membranes this occurs, and whether this is indeed specific to PA. Thus, although there is clear support for a role of Mic19 in mitochondrial lipid metabolism and physiological consequences in an in vivo model, the mechanistic explanation for this is still lacking." The new data on liposome extraction and PA transfer is nice but lacks important controls. Another His-tagged protein as control would be needed and also other lipids should be tested. As it is now, the transfer is not shown to be specific to PA. Moreover, the authors still do not show in which compartments (ER, MAM, OM, IM) which lipids accumulate. We agree that this is tricky but to make this claim of specific PA transport it is essential.

As the rest of the paper is really interesting (in particular the mouse data) and of high quality I strongly to either improve this point or to reformulate this claim. A general role of MIC19/SLC25A46 in lipid metabolism is of sufficient general interest. My suggestion is that the authors should heavily downsize their claim. The authors should rather focus on the general role in lipid metabolism. They should remove the scheme in Fig 4L as they have no data to show transport of PA from ER to mitochondria. They might have some circumstantial data, but many pieces are missing to prove this scheme.

The authors themselves think that they cannot differentiate whether the effects are coming from MIC19 or other MICOS components and that it is possible that MIC19 effects could be indirect. Also because of this, I would recommend to downscale the claims of MIC19 role in PA transport.

Reviewer #3:

Remarks to the Author:

The authors have provided additional data to strengthen the mechanisms they propose, particularly in regard to PA transport, as well as CLS knockdown which demonstrated that the loss of MICOS is probably CL dependent. However, they have still not mechanistically connected the observations that result from Mic19 loss to one another, to be able to put them into a pathway. The manuscript could still be improved by any rescue experiment other than Mic19 reexpression that rescues downstream phenotypes. That said, the authors have revised the manuscript to less directly make these claims. Some sections of the paper could still be revised to make clear that while each of these phenotypes result from Mic19 loss, their relationship(s) have not been fully tested in this model. However, overall the manuscript is improved.

Reviewer #4:

Remarks to the Author:

In this manuscript, the authors focus on the role of Mic19 and SLC25A46 on PA binding and transport. However, lipidomics data show that not only PA and CL, but also PC, PS and PE lipids decrease in mitochondrial fraction from livers of Mic19 LKO mice, suggesting a severe effect on lipid homeostasis that could be related to destabilization of the MICOS complex and loss of mitochondrial cristae, as during loss of SLC25A46 (previously published, PMID 27390132). So, this problem does not appear to be specifically related to PA. Furthermore, the dissociation of contact sites between ER and mitochondria and the destabilization of MICOS could have an effect on other lipid-transfer proteins that specifically or non-specifically transport PA, without Mic19 and

SLC25A46 being directly involved in PA transfer between membranes. This is somehow confirmed by the rather unconvincing data provided by the authors. Indeed, lipid transfer assays show that Mic19 and SLC25A46 promote PA transfer by barely a factor of 2 compared with the control (no protein), suggesting that the direct contribution of these proteins to PA transfer is in fact extremely poor. Unfortunately, the authors' conclusions do not substantiate the fact that these proteins have a very low propensity to transfer PA by themselves.

It is also extremely unclear why the authors associate disordered lipid binding regions with any specific binding to PA. While it is true that some unstructured regions do bind to membranes, these bindings are hardly ever specific to a particular lipid; instead, these regions recognize membrane physico-chemical properties. Moreover, these regions are not known to extract and transfer lipids between membranes. Many lipid-transfer proteins contain unstructured regions, but also well-folded domains responsible for lipid solubilization and transfer. See recent reviews on this (e.g. PMID 37680133). What's most surprising about the demonstration is that the largest region of Mic19 predicted to be a disordered lipid binding region coincides with a coiled-coil region, illustrating the limitations of these prediction programs. In conclusion, the arguments for PA-specific binding and transfer by Mic19 and SLC25A46 are hardly convincing, making this part of the manuscript too weak at this stage. I'm pointing out another problem here, but related to the previous one: The authors purify and study SLC25A46 as if it was a soluble protein, whereas it contains one, or even several, transmembrane domains. There is no evidence that the protein is well folded after the purification steps (no structural analysis, gel filtration etc.). The authors indicate in figure S4A that SLC25A46 has one TM domain, whereas other studies indicate that this protein may contain up to 6 transmembrane domains (PMID 27390132; 30178502; see also AlphaFold prediction). Manipulating this protein in solution makes little sense, a point I made in my previous review, but which was not taken into account by the authors. Given the data provided so far, I suggest that the authors take a more structural/molecular approach if they want to prove that Mic19 and SLC25A46 are PA-binding and PA-transfer proteins. If not, they need to seriously reconsider their claims.

We sincerely thank the reviewers for their valuable comments and constructive criticisms. Based on the comments and suggestions of reviewers and editor, we deleted some unrated data or words and re-organized the manuscript. Reviewers' comments are responded point-by-point as below. Reviewers' points are underlined and italic for easier reference. Descriptions for all newly performed experimental results and other changes (compared to manuscript-R1) are highlighted in blue in the manuscript-R2.

Reviewer #1

The authors addressed my concerns. I suggest moving Figure S9G-J to main Figure 9.

We appreciate the reviewer's comments and suggestions. According to the reviewer's comments and editor's suggestions, we deleted Figure 4 and S4 of the manuscript-R1 and the related words in the revised manuscript-R2, and moved Figure S9G-J to main Figure 8 (Figure 9 in the Manuscript-R1) of the revised manuscript-R2.

Reviewer #2

The authors have performed some additional experiments of which some are convincing, yet the major concern has not been resolved sufficiently in my opinion. The claim about the role of MIC19 in PA transport is grossly overstated. As we previously said, we still have to say that there is no sufficient "...biochemical evidence that Mic19 or SLC25A46 directly mediate the transport of PA, or between which membranes this occurs, and whether this is indeed specific to PA. Thus, although there is clear support for a role of Mic19 in mitochondrial lipid metabolism and physiological consequences in an in vivo model, the mechanistic explanation for this is still lacking." The new data on liposome extraction and PA transfer is nice but lacks important controls. Another His-tagged protein as control would be needed and also other lipids should be tested. As it is now, the transfer is not shown to be specific to PA. Moreover, the authors still do not show in which compartments (ER, MAM, OM, IM) which lipids accumulate. We agree that this is tricky but to make this claim of specific PA transport it is essential.

We sincerely thank the reviewer for his/her comments. We agree with the reviewer's comment that our manuscript lacks direct biochemical evidence of Mic19 or SLC25A46 in PA transport. In addition, it is hard to obtain the purified fractions of ER, MAM, OM, IM. Therefore, according to the reviewer's comments and editor's suggestions, we deleted Figure 4 and S4 of

the manuscript-R1 and the related words in the revised manuscript-R2. The data about the role of Mic19 or SLC25A46 in PA transport will be showed and further explored in our future study.

As the rest of the paper is really interesting (in particular the mouse data) and of high quality I strongly to either improve this point or to reformulate this claim. A general role of MIC19/SLC25A46 in lipid metabolism is of sufficient general interest. My suggestion is that the authors should heavily downsize their claim. The authors should rather focus on the general role in lipid metabolism. They should remove the scheme in Fig 4L as they have no data to show transport of PA from ER to mitochondria. They might have some circumstantial data, but many pieces are missing to prove this scheme.

The authors themselves think that they cannot differentiate whether the effects are coming from MIC19 or other MICOS components and that it is possible that MIC19 effects could be indirect. Also because of this, I would recommend to downscale the claims of MIC19 role in PA transport.

We sincerely thank the reviewer's comments and suggestions. According to the reviewer's comments and editor's suggestions, we have deleted Figure 4 and S4 of the manuscript-R1 and the related claims in the revised manuscript-R2.

Reviewer #3

The authors have provided additional data to strengthen the mechanisms they propose, particularly in regard to PA transport, as well as CLS knockdown which demonstrated that the loss of MICOS is probably CL dependent. However, they have still not mechanistically connected the observations that result from Mic19 loss to one another, to be able to put them into a pathway. The manuscript could still be improved by any rescue experiment other than Mic19 reexpression that rescues downstream phenotypes. That said, the authors have revised the manuscript to less directly make these claims. Some sections of the paper could still be revised to make clear that while each of these phenotypes result from Mic19 loss, their relationship(s) have not been fully tested in this model. However, overall the manuscript is improved.

We appreciate the reviewer's comments and suggestions. According to the reviewer's comments and editor's suggestions, we have deleted the data about PA transport (Figure 4, S4 and the related claims of the manuscript-R1) in the revised manuscript-R2. The data about the role of Mic19 or SLC25A46 in PA transport will be showed and further explored in our future study.

Reviewer #4

In this manuscript, the authors focus on the role of Mic19 and SLC25A46 on PA binding and transport. However, lipidomics data show that not only PA and CL, but also PC, PS and PE lipids decrease in mitochondrial fraction from livers of Mic19 LKO mice, suggesting a severe effect on lipid homeostasis that could be related to destabilization of the MICOS complex and loss of mitochondrial cristae, as during loss of SLC25A46 (previously published, PMID 27390132). So, this problem does not appear to be specifically related to PA. Furthermore, the dissociation of contact sites between ER and mitochondria and the destabilization of MICOS could have an effect on other lipid-transfer proteins that specifically or non-specifically transport PA, without Mic19 and SLC25A46 being directly involved in PA transfer between membranes. This is somehow confirmed by the rather unconvincing data provided by the authors. Indeed, lipid transfer assays show that Mic19 and SLC25A46 promote PA transfer by barely a factor of 2 compared with the control (no protein), suggesting that the direct contribution of these proteins to PA transfer is in fact extremely poor. Unfortunately, the authors' conclusions do not substantiate the fact that these proteins have a very low propensity to transfer PA by themselves.

We sincerely thank the reviewer's comments and suggestions. According to the reviewer's comments and editor's suggestions, we have deleted the data about PA transport (Figure 4, S4 and the related claims of the manuscript-R1) in the revised manuscript-R2. The data about the role of Mic19 or SLC25A46 in PA transport will be showed and further explored in our future study.

It is also extremely unclear why the authors associate disordered lipid binding regions with any specific binding to PA. While it is true that some unstructured regions do bind to membranes, these bindings are hardly ever specific to a particular lipid; instead, these regions recognize membrane physico-chemical properties. Moreover, these regions are not known to extract and transfer lipids between membranes. Many lipid-transfer proteins contain unstructured regions, but also well-folded domains responsible for lipid solubilization and transfer. See recent reviews on this (e.g. PMID 37680133). What's most surprising about the demonstration is that the largest region of Mic19 predicted to be a disordered lipid binding region coincides with a coiled-coil region, illustrating the limitations of these prediction programs. In conclusion, the arguments for PA-specific binding and transfer by Mic19 and SLC25A46 are hardly convincing.

making this part of the manuscript too weak at this stage.

We appreciate the reviewer's comments and suggestions. We agree with the reviewer's comment that our manuscript lacks direct biochemical evidence of Mic19 or SLC25A46 in PA transport. According to the reviewer's comments and editor's suggestions, we have deleted the data about PA binding and transport (Figure 4, S4 and the related claims of the manuscript-R1) in the revised manuscript-R2. The role of Mic19 or SLC25A46 in PA binding and transport will be further explored in our future study.

I'm pointing out another problem here, but related to the previous one: The authors purify and study SLC25A46 as if it was a soluble protein, whereas it contains one, or even several, transmembrane domains. There is no evidence that the protein is well folded after the purification steps (no structural analysis, gel filtration etc.). The authors indicate in figure S4A that SLC25A46 has one TM domain, whereas other studies indicate that this protein may contain up to 6 transmembrane domains (PMID 27390132; 30178502; see also AlphaFold prediction). Manipulating this protein in solution makes little sense, a point I made in my previous review, but which was not taken into account by the authors. Given the data provided so far, I suggest that the authors take a more structural/molecular approach if they want to prove that Mic19 and SLC25A46 are PA-binding and PA-transfer proteins. If not, they need to seriously reconsider their claims.

We sincerely thank the reviewer's comments and suggestions. We also found that SLC25A46 may contain different number of TM domain by prediction with the different software. According to the reviewer's comments and editor's suggestions, we have deleted the data about the function of SLC25A46 in PA binding and transport (Figure 4, S4 and the related claims of the manuscript-R1) in the revised manuscript-R2. The role of Mic19 or SLC25A46 in PA binding and transport will be further explored in our future study.